# Towards Large-Scale In-Context Reinforcement Learning by Meta-Training in Randomized Worlds

*†Fan Wang[1,2], *Pengtao Shao[2], Yiming Zhang[2], Bo Yu[2], †Shaoshan Liu[2],
Ning Ding[2], Yang Cao[1,3], Yu Kang[1,3], and Haifeng Wang[4]

[1]University of Science and Technology of China, Heifei, China
[2]Shenzhen Institute of Artificial Intelligence and Robotics for Society, Shenzhen, China
[3]Anhui Province Key Laboratory of Intelligent Low-Carbon Information Technology and Equipment
[4]Baidu Inc, Beijing, China

## Abstract

In-Context Reinforcement Learning (ICRL) enables agents to learn automatically and on-the-fly from their interactive experiences. However, a major challenge in scaling up ICRL is the lack of scalable task collections. To address this, we propose the procedurally generated tabular Markov Decision Processes, named AnyMDP♣. Through a carefully designed randomization process, AnyMDP is capable of generating high-quality tasks on a large scale while maintaining relatively low structural biases. To facilitate efficient meta-training at scale, we further introduce decoupled policy distillation and induce prior information in the ICRL framework♠. Our results demonstrate that, with a sufficiently large scale of AnyMDP tasks, the proposed model can generalize to tasks that were not considered in the training set through versatile in-context learning paradigms. The scalable task set provided by AnyMDP also enables a more thorough empirical investigation of the relationship between data distribution and ICRL performance. We further show that the generalization of ICRL potentially comes at the cost of increased task diversity and longer adaptation periods. This finding carries critical implications for scaling robust ICRL capabilities, highlighting the necessity of diverse and extensive task design, and prioritizing asymptotic performance over few-shot adaptation.

## 1 Introduction

In-Context Learning (ICL) [1] has emerged as a pivotal paradigm for large pre-trained models [2–5], enabling adaptation to novel tasks without parameter adjustments. Unlike gradient-based in-weight learning (IWL), which modifies parameter weights through optimization, ICL facilitates skill acquisition through natural interaction. This is achieved by unifying versatile learning paradigms by sequence of interactions in the context, including supervised learning [6, 7], imitation learning [8–10], and reinforcement learning [11–13], etc. By leveraging those contexts, ICL eliminates reliance on manually engineered objective functions and the labor-intensive optimization infrastructure required by traditional IWL frameworks, providing versatile, self-directed manners of learning. ICL further aligns with black-box meta-learning principles [14, 11] where task-agnostic adaptation occurs through computation and memorization rather than explicit parameter tuning. It can be regarded as a unifying

---

*Equal Contribution
†Corresponding to: fanwang.px@gmail.com, shaoshanliu@cuhk.edu.cn

♣`https://github.com/FutureAGI/Xenoverse/tree/main/xenoverse/anymdp`
♠`https://github.com/airs-cuhk/airsoul/tree/main/projects/OmniRL`

39th Conference on Neural Information Processing Systems (NeurIPS 2025).

framework that bridges the gap between conventional meta-learning approaches and contemporary large-scale language model capabilities.

The historical conception of ICL has been predominantly associated with supervised few-shot learning paradigms. Recently, the frameworks for learning from dynamic experiences have achieved growing success, where the contextual information can encompass observations, actions, feedback, and reasoning. Among these emerging paradigms, In-Context Reinforcement Learning (ICRL) stands out as a critical approach. Unlike traditional ICL, which arises from self-supervised pre-training, ICRL typically requires explicit training via reinforcement learning (RL) algorithms [12], introducing substantially higher computational complexity and training costs for large-scale applications. While recent supervised learning approaches have shown promise in enhancing ICRL efficiency [13, 15, 16], current implementations remain constrained by limited context lengths (typically <1K tokens) and narrow task scales (often <1K distinct tasks). These limitations significantly restrict ICRL's capacity for broad task generalization, highlighting an urgent need for scalable architectures that can reconcile contextual flexibility with computational efficiency.

Additionally, existing benchmarks for ICRL predominantly fall into two categories: 1) oversimplified environments like multi-armed bandit tasks, or 2) procedurally generated seed tasks such as maze navigation [17–19] or arcade-style games [20]. While the former lacks practical relevance due to their constrained problem spaces, the latter suffers from superficial parameter randomization - diversifying only peripheral elements like visual textures or sub-objectives while preserving core environmental biases (e.g., fixed transition dynamics and reward functions). This design paradigm confines ICRL agents to narrow problem distributions, yielding systems that demonstrate limited generalization scope. Consequently, scaling task complexity beyond a few hundred variants encounters diminishing returns, as models inherit fundamental limitations from their training environments' structural biases.

To advance the scalability and generalization capabilities of ICRL, this work presents two interrelated contributions: a novel benchmarking task set and a scalable ICRL framework. First, we introduce AnyMDP, a scalable task generation environment where Markov Decision Processes (MDPs) are systematically designed with fully randomized transition dynamics and reward functions. To balance environmental richness with learner challenge, we propose the procedural generation process remarked with banded transition matrices. This design minimizes structural biases while preserving problem complexity, creating a benchmark that demands genuine contextual adaptation rather than exploiting environmental regularities. Second, we propose Decoupled Policy Distillation (DPD) to enhance ICRL training efficiency. By disentangling reference policy from behavior policy, and inducing prior knowledge into the context, we achieve unprecedented training scales and ICRL versatility. Our implementation, OmniRL, was trained exclusively on AnyMDP tasks using context sequences of up to 512K steps per sequence and 6 billion steps overall. Empirical validation demonstrates OmniRL's capability of many-shot generalization to entirely novel tasks, confirming our models generalization scope and versatile learning ability.

Moreover, leveraging the scalable AnyMDP task set, we conduct ablations that reveal critical insights into ICRL's dependence on *task diversity* and *long-context modeling*. First, the boundary between IWL and ICL is fundamentally determined by training-task coverage: sufficiently diverse distributions (at least 10K unique tasks in our case) elicit emergent task learning capabilities, whereas limited coverage reduces systems to task recognition regimes. Second, long-context modeling proves indispensable for interpreting complex task specifications and achieving broad generalization. These findings establish a new paradigm for ICRL research in which scalable benchmarks and context-aware training jointly yield systems that approach human-like adaptability in dynamic decision-making domains.

## 2 Related Work

### 2.1 Emergence of In-Context Learning

Meta-learning, also known as learning to learn [21, 22, 12], pertains to a category of approaches that prioritize the acquisition of generalizable adaptation skills across a spectrum of tasks. Large models pre-trained on massive, uncurated datasets naturally give rise to In-Context Learning (ICL), a phenomenon analogous to model-based meta-learning [1, 8, 23–27]. Investigations reveal that ICL ability is tightly linked to pre-training data distribution, with factors spanning burstiness and intra-sequence correlation [28, 29, 25] to overall diversity [30]. Complementary analyses show

that computation-based ICL can emulate a rich spectrum of learning behaviors, including gradient descent [31–33], Bayesian inference [33], and broader learning paradigms [13]. Those findings suggest its potential as a general-purpose learning machine [34–37] rather than a mere mechanism for instruction following or few-shot adaptation. Building on the recent progress in ICL and the principles of meta-learning, we adopt the term meta-training to denote the training procedure that explicitly equips the model with the capacity to learn in context, thereby distinguishing it from pre-training, which yields only zero-shot skills and incentivizes ICL as an incidental by-product of exposure to uncurated data.

## 2.2 In-Context Reinforcement Learning

In-context reinforcement learning (ICRL) encompasses algorithms that dynamically adapt to decision-making tasks by synthesizing self-generated trajectories and incorporating external feedback in the context [12, 13]. Unlike one-off ICL adaptation, ICRL stresses continual policy improvement driven by accumulated experience, echoing the "third system" forged through ongoing interaction and experience accumulation, as articulated by Barabasi et al. [38]. ICRL typically employs recurrent [12, 39] and attention-based [13, 40] neural structures that are capable of encoding the interactive histories in the *inner loop*. The meta-training process that searches the parameters for learning functionality is called *outer loop*. Common choices for the outer-loop optimizer for ICRL include reinforcement learning [12, 41, 40], evolutionary strategies [42, 17], and supervised learning [13, 16]. While supervised learning generally achieves higher sample efficiency compared to RL and evolutionary strategies, it often suffers from the bottleneck of distribution shift [43, 44].Additionally, the absence of an oracle policy can become a critical bottleneck for supervised learning; consequently, alternatives replace the expert with an RL coach for data synthesis [8, 45]. A further obstacle to ICRL research is the lack of large-scale task suites, which hampers the emergence of ICRL in real-world decision-making problems and motivates the creation of procedurally generated tasks at scale.

## 2.3 Procedurally Generated Tasks

Commonly employed procedural-generation techniques include randomizing a subset of domain parameters to create variant tasks by randomizing the rewards or targets while keeping the transitions fixed [13, 16, 40, 22, 46, 47], randomizing the dynamics while keeping the targets unchanged [42, 17, 20, 48, 49, 18], and randomizing the observations and labels without altering the underlying transitions and rewards [34, 19, 50]. These techniques is frequently applied to enhance the robustness of decision and policy model for sim-to-real transfer [51–53], namely domain randomization (DR). Although DR creates a class of tasks with a certain variety, it is restricted by the original task setting and basically forms a close and finite task set. Recently, generating open-ended tasks with enhanced diversity has emerged as a focal point in research, with a significant emphasis on game-based frameworks [54, 52, 36, 55, 54, 52, 56]. Such approaches are increasingly recognized as vital inductive biases for fostering adaptability. While procedurally generated discrete Markov Decision Processes (MDPs) provide a promising avenue for minimizing structural bias, owing to their configuration via a constrained set of hyperparameters, naive sampling of these MDPs often results in a concentration of trivial tasks, which lack meaningful complexity [57, 58]. In this work, we address this limitation by strategically incorporating critical features during the procedural generation of discrete MDPs.

# 3 AnyMDP: Procedural Generation of High-Quality MDP Tasks

## 3.1 Problem Setting and Definitions

We denote a task of Markov Decision Processes (MDPs) with $\tau = \langle \mathcal{S}, \mathcal{A}, \mathcal{P}, \mathcal{R}, \mathcal{S}_0, \mathcal{S}_E, T_E, \gamma \rangle$ where $\mathcal{S}$ denotes the state space, $\mathcal{A}$ the action space, $\mathcal{P} : \mathcal{S} \times \mathcal{A} \to \mathcal{S}$ the state-transition probability distribution, $\mathcal{R} : \mathcal{S} \times \mathcal{A} \times \mathcal{S} \to \mathbb{R}$ the reward function, $\mathcal{S}_0$ the set of initial states, and $\mathcal{S}_E$ the set of terminal states, $T_E$ the maximum steps in an episode, and $\gamma$ the discount factor. We consider only fully visible discrete MDPs with $\mathcal{S} = \{1, ... n_s\}$, $\mathcal{A} = \{1, ... n_a\}$, denoted by $\tau$. We define average state transition $P_{\mathfrak{r}}(s'|s) = \mathbb{E}_{a \sim \pi^{\mathfrak{r}}(a|s)|}$, which is the transition under uniform policy $\mathfrak{r} : a \sim \pi^{\mathfrak{r}}(a|s) = \frac{1}{|\mathcal{A}|}$. For absorbing states $s \in \mathcal{S}_E$, the transition was connected back to $\mathcal{S}_0$. We then formally define the *AnyMDP* task collection $\mathcal{T}(n_s, n_a, \eta, \epsilon, \kappa, b_-, b_+)$ as MDPs satisfying the following conditions:

- *Ergodic*: average state transition $P_\mathfrak{r}$ has unique positive stationary distribution.
- *Banded Transition Matrix*: there exists a ranking of states: $s_1, ..., s_{n_s}$, such that: $P_\mathfrak{r}$ satisfies the following condition:

$$P_\mathfrak{r}(s_j|s_i) \equiv 0, \forall j < i - b_-, \text{ or } j > i - b_+$$

$$\sum_{j<i} P_\mathfrak{r}(s_j|s_i) > \eta, \text{ if } i > b_-, \sum_{j>i} P_\mathfrak{r}(s_j|s_i) > \epsilon, \text{ if } i < n_s - 1 \qquad (1)$$

- *Ascending Value Function*: given the aforementioned ranking of states, the value function $V^*$ under the optimal policy satisfies the following condition:

$$V^*(s_{n_s}) > \max_{s \in \mathcal{S}_0} V^*(s) + \kappa \qquad (2)$$

Notice that $\mathcal{T}(n_s = 1, n_a)$ represents the widely used multi-armed bandits benchmark for ICRL [12, 16, 40]. As the number of states $n_s$ increases, tasks demand progressively more sophisticated reasoning over delayed rewards and complex state spaces. This transition evolves the problem from a simple multi-armed bandit framework to a full reinforcement learning paradigm, where long-term planning and environmental interaction become critical. Furthermore, since the ground truth dynamics $\mathcal{P}$ and reward function $\mathcal{R}$ are known within the simulation environment, this setup enables straightforward computation of the oracle solution through value iteration [59].

## 3.2 Motivations of AnyMDP

AnyMDP is motivated by the following visions: (1) Existing procedurally generated MDP benchmarks like PROCON [57] and Garnet [58] overlook terminal states ($\mathcal{S}_E$) which is a critical feature of real-world environments where agents enter terminal states (e.g., task completion or failure). Therefore, AnyMDP allows terminal states to match many real tasks. (2) Through banded transition matrix and ascending value function, AnyMDP has high-valued states that are exponentially less possible to reach by random exploration. In real-world tasks, especially long-horizon ones, the probability of reaching the goal by naïve random choices decays exponentially with task length. Consequently, solely relying on random exploration can be extremely inefficient in solving those tasks; instead, a continuous exploration–exploitation trade-off is required to approach the optimum gradually. Inspired by this necessity, we impose on the Markov Decision Process the constraint that *states with higher values are reached with lower probability*, thereby guiding the learner toward systematic, intelligent exploration rather than mere random search.

By defining the stationary distribution $p_\mathfrak{r}$ with $p_\mathfrak{r} P_\mathfrak{r} = p_\mathfrak{r}$, the upper and lower bound of the chances of arriving at any states is ensured by the following theorem:

**Theorem 1.** *Given the condition of Equation* (1)*, for $j > b$, where $b = \max(b_-, b_0) + b_+$, there exists a value $0 < \delta < 1/(b_+ + 1)$. If $\eta > 1 - \delta$, the sampling algorithm ensures that the transition $\hat{\mathcal{P}}_\mathfrak{r}$ has a unique positive stationary distribution (Ergodic). Specifically, for the state $s_j$, the $p_\mathfrak{r}(s_j)$ satisfies the following bounds:*

$$C_1 \epsilon^{j-b} < p_\mathfrak{r}(s_j) < C_2(1 - \delta)^{j-b},$$

*where $C_1$ and $C_2$ are positive constants.*

Theorem 1 guarantees that the probability of staying at $s_j$ by random choice of actions decreases at least exponentially with respect to $j$, while also being bounded above by another exponentially decaying, yet non-negligible, probability to ensure ergodicity. Using a banded transition matrix, we further equip the MDP with a *Composite Reward* (CR) that decomposes reward generation into independent components, ensuring an ascending value function while preserving low structural bias when sampling. We have relegated the proof of Theorem 1 and the details of the sampling procedures to the Appendix B for brevity.

## 3.3 Comparison with Other Procedural MDPs and Empirical Validation

The proposed sampler guarantees high-quality MDPs in three aspects: ergodicity, low structural bias, and high learning difficulty. To empirically verify the high learning difficulty and to validate

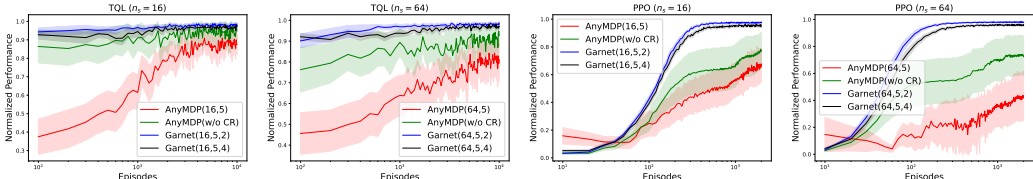

Figure 1: An ablation study comparing AnyMDP tasks with Garnet MDP and AnyMDP without composite reward demonstrates that the procedural generation algorithm of AnyMDP produces tasks of higher learning difficulty.

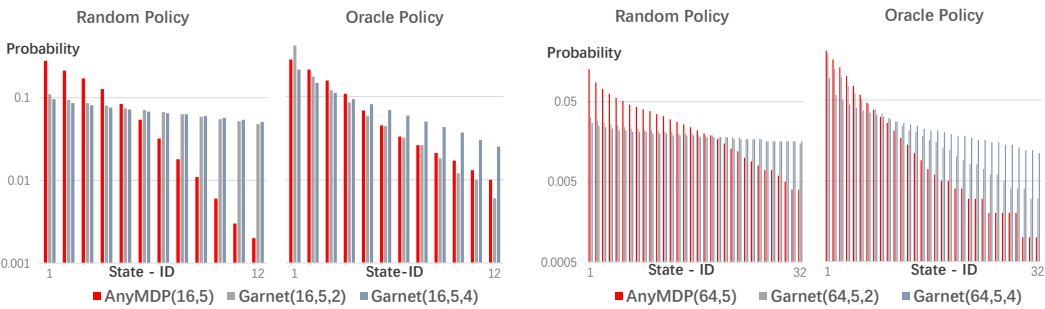

Figure 2: Comparison of the stationary distributions (SDs) of the oracle policy and the uniform random policy across four classes of environments: AnyMDP - $\tau(16/64, 5)$, and Garnet MDP-$(16/64, 5, 2/4)$ demonstrates that AnyMDP exhibits uniquely exponentially-decaying SDs. The results are averaged over 64 randomly sampled tasks, with the SDs for each task re-ranked in descending order.

Theorem 1, we run two groups of experiments. Figure 1 compares the performance of two well-known discrete MDP learning methods: Tabular Q-Learning [60] with Upper Confidence Bound (TQL-UCB) [61] and Proximal Policy Optimization (PPO) [62], on tasks sampled from AnyMDP and Garnet MDP [58]. Garnet uses parameters $n_s, n_a, b, \sigma, \tau$ to shape generated MDPs. Here, we set $\sigma = 0.1, \tau = 0, b = \{2, 4\}$, and keeps $n_s$ and $n_a$ equal to AnyMDP for comparison. To isolate the impact of composite reward (CR) Equation (10) sampling, we also include a baseline without the composite reward sampling technique (AnyMDP w/o CR). The results show that AnyMDP tasks pose greater challenges for both RL methods. Figure 2 further illustrates the stationary distribution (SD) of states by tasks sampled from AnyMDP and Garnet MDPs. It demonstrates exponential decay in SD of AnyMDP, empirically validating Theorem 1, while Garnet MDPs exhibit flatter SDs, especially under the random policy.

## 4   The Scalable ICRL Framework of OmniRL

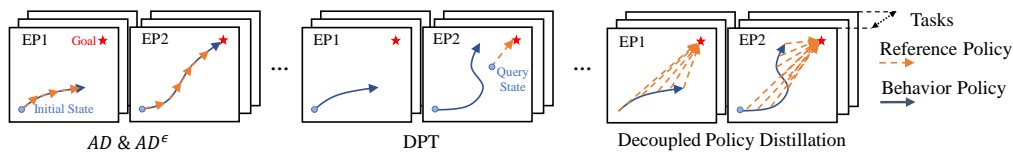

Figure 3: Comparison of the learning pipelines and data formulation of different ICRL methods (AD, $AD^\varepsilon$, DPT) and Decoupled Policy Distillation (DPD).

**Decoupled Policy Distillation (DPD)**: Meta-training for ICRL using RL or Evolution Strategies faces significant costs, including cumbersome infrastructure requirements and high computational costs. Recent advances in supervised learning-based meta-training methods—such as Algorithm Distillation (AD) [13], $AD^\varepsilon$ [15], and Decision Pre-Training Transformers (DPT) [16]—have shown promise for scalable ICRL meta-training. However, a critical challenge arises during inference: contextual trajectories are generated by the model itself, creating an unavoidable gap between

training-time and inference-time trajectories. This discrepancy can lead to catastrophic failures during deployment. While leveraging diverse behavior trajectories has been shown to mitigate this issue [16], it introduces a new challenge of maintaining training efficiency. To address these limitations, we propose Decoupled Policy Distillation (DPD), a framework inspired by DPT and data aggregation techniques for imitation learning [63, 64]. Our approach hinges on two decoupled policies: The *behavior policy* refers to the policy executed to generate trajectories during training. The *reference policy* refers to the target policy to be imitated, which remains decoupled from direct execution. Decoupling the behavior and reference policies enables the introduction of diversity into the behavior policy, thereby reducing the discrepancy between training and inference trajectories while maintaining the optimality of the reference policy, as shown in Figure 3. Unlike DPT, which imitates only one-step action conditioned on a trajectory, our step-wise supervision framework is inherently designed to align with high-efficiency chunk-wise training pipelines for sequence models such as Transformers [65] and their optimized variants [66, 67], producing $T$ times training efficiency ($T$ is the sequence length).

**Prior knowledge augmented ICRL**: For SS, which employs diverse policies for trajectory generation, prior knowledge becomes crucial for interpreting actions derived from heterogeneous policies [68]. This motivates the incorporation of prior knowledge, specifically metadata indicating the policy used to generate each action. In this work, we implement a diverse set of behavior policies $\Pi = \{\pi^{(b)}\}$, with $(b)$ denoting the behavior policies. $\pi^{(b)}$ comprises seven distinct types, including myopic greedy, oracle, Q-learning, and model-based reinforcement learning, denoted by a marker $tag(\pi) \in \{0, ..., 6\}$. To handle unseen or unclassified policies, we reserve an additional identifier "Unk" with $m = 7$. The trajectory is denoted by $h_T(\tau, \pi) = [(s_1, g_1, a_1, r_1), ..., (s_t, g_T, a_T, r_T)]$, where $s_t \in \mathcal{S}$, $a_t \in \mathcal{A}$, $a_t \sim \pi^{(b)}(s_t)$, $r_t \sim \mathcal{R}(s_t, a_t, s_{t+1})$, and $g_t = g(a_t) = tag(\pi^{(b)})$ denotes the prior knowledge for action.

**Chunwise Training**: We use $\pi^*$ to denote the reference policy, which is the oracle policy with $\gamma > 0.99$. It is then used to label a list of the reference actions step-by-step as $l_T(\tau) = [a_1^*, a_2^*, ..., a_T^*]$ with $a_t^* \sim p_t^* = \pi^*(s_t)$. Notice that for most of the time $a_t \neq a_t^*$. We first collect the training and validation datasets by:

$$\mathcal{D}(\mathcal{T}) = \{< h_T(\tau, \pi), l_T(\tau) > | \tau \sim \mathcal{T}, \pi \sim \Pi\} \tag{3}$$

Scaling ICL to handle complex tasks necessitates the efficient modeling of extensive contexts, which in turn demands substantial computational resources and large-capacity hardware memories. To further break down the limitation in context length, we break a long sequence $h_T$ into K segments $[1, T_1], [T_1 + 1, T_2], ..., [T_{K-1} + 1, T_K]$. The forward pass is calculated recurrently across the segments, and the backward calculation is performed within each segment. The gradient for the memory states of the linear attention layer $\phi_t$ is blocked across the segments. Thus the sequence is encoded in the following chunkwise manner:

$$p_{T_k+1}^\theta, ..., p_{T_{k+1}}^\theta, \phi_{k+1} = Causal_\theta(SG(\phi_k), s_{T_k+1}, ..., s_{T_k+2}, ..., ..., s_{T_{k+1}}) \tag{4}$$

with $SG$ representing stopping gradient. In the meta-training process, the gradients are calculated within each segment and accumulated in cache first. They are applied to the parameters only at the end of the trajectory. The final target is as follows:

$$Minimize : \mathcal{L} \propto - \sum_{h_T, l_T \in \mathcal{D}} \sum_t w_t log p_t^\theta(a_t^*) \tag{5}$$

The complete architecture of the model and its training pipeline are depicted in Figure 4. We utilize a causal sequence model (where the prediction at each position relies solely on information from preceding positions) to encode the input sequence and forecast the action at the position aligned with the state inputs. In this paper, we mainly employ linear attention layers including gated slot attention (GSA) [67] layers, mamba-2 [69], and RWKV-7 [70] as the causal sequence models.

# 5 Experiments

## 5.1 Demonstration of Generalization and Scalability

We first validate the representational capability of AnyMDP tasks as universal MDPs. To this end, we collect a dataset $\mathcal{D}_{tra}(\mathcal{T}(n_s, n_a))$ comprising $512K$ sequences for training, where $n_s \in$

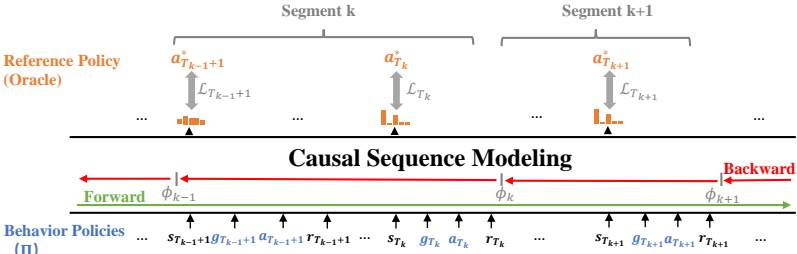

Figure 4: OmniRL model structure and training losses.

Table 1: Comparison of best average episodic performance within 10,000 episodes for each learner. The episode performances are normalized to a scale of $0\%$ (uniform random policy) to $100\%$ (oracle policy). The minimum steps and episodes required to achieve within at least $99\%$ of the best episodic performances are also listed. The hyper-parameters for Q-Learning and PPO are optimized under the evaluated task or task set. Results for AnyMDP and Garnet Tasks are averaged over $64$ tasks, results for Gymnasium tasks are averaged over $3$ independent runs for each task. Gymnasium, DarkRoom, and Bandits tasks are entirely absent from OmniRL's training regimen.

| Environments | Performances / AVG. *Steps* cost / AVG. *Episodes* cost | | | |
| --- | --- | --- | --- | --- |
| | **TQL-UCB** | **PPO** | **OmniRL** (AnyMDP) | **OmniRL** (GarnetMDP) |
| $\mathcal{T}_{tst}(1,5)$ (Bandits) | $92.1\%/100/100$ | $95.6\%/1.2K/1.2K$ | $82.5\%/103/103$ | $46.6\%/33/33$ |
| $\mathcal{T}_{tst}(16,5)$ | $92.0\%/297K/4.7K$ | $90.6\%/476K/9.7K$ | $95.3\%/2.0K/29$ | $47.8\%/1.6K/24$ |
| $\mathcal{T}_{tst}(32,5)$ | $84.7\%/616K/5.6K$ | $72.2\%/618K/9.7K$ | $90.3\%/6.5K/47$ | $42.0\%/5.0K/44$ |
| $\mathcal{T}_{tst}(64,5)$ | $83.7\%/1.1M/5.1K$ | $58.3\%/1.1M/9.4K$ | $91.3\%/7.7K/25$ | $47.1\%/6.6K/24$ |
| $\mathcal{T}_{tst}(128,5)$ | $73.2\%/1.8M/6.9K$ | $49.0\%/1.3M/8.6K$ | $80.2\%/36.3K/100$ | $32.3\%/9.0K/31$ |
| Garnet$(16,5,2)$ | $98.8\%/241K/2.1K$ | $97.1\%/57K/0.5K$ | $85.9\%/8.2K/71$ | $99.0\%/10.8K/95$ |
| Garnet$(64,5,2)$ | $98.7\%/614K/1.7K$ | $98.1\%/96K/0.26K$ | $80.4\%/8.0K/19$ | $87.3\%/7.4K/23$ |
| CliffWalking | $100\%/3.1K/35$ | $95.9\%/99.3K/2.7K$ | $100\%/3.0K/65$ | $63\%/29K/300$ |
| FrozenLake (non-slippery) | $95.3\%/23.6K/3.7K$ | $96.8\%/18.2K/2.1K$ | $99.8\%/0.3K/35$ | $75.1\%/4.0K/250$ |
| FrozenLake (slippery) | $96\%/208K/10.0K$ | $95.6\%/73.6K/4.7K$ | $79.5\%/7.7K/245$ | $31.3\%/11.8K/800$ |
| Discrete-Pendulum (g=1) | $94.9\%/22K/110$ | $99.3\%/198K/990$ | $90.5\%/8K/40$ | $0.0\%/-/-$ |
| Discrete-Pendulum (g=5) | $99.7\%/426K/2.13K$ | $99.8\%/132K/660$ | $91.8\%/34K/170$ | $0.0\%/-/-$ |
| Discrete-Pendulum (g=9.8) | $90.2\%/2.0M/10.0K$ | $98.3\%/186K/930$ | $73.4\%/33K/165$ | $0.0\%/-/-$ |
| Switch2 (Multi-Agent)[71] | $98\%/3.8K/110$ | $-$ | $80.4\%/2.8K/100$ | $-$ |
| Darkroom (6x6) | $98.1\%/6.2K/481$ | $97.6\%/10.6K/560$ | $95.2\%/845/40$ | $90.5\%/21.3K/440$ |
| Darkroom (8x8) | $96.8\%/24.5K/2.0K$ | $96.7\%/15.9K/930$ | $93.8\%/1.5K/40$ | $88.9\%/30.4K/480$ |
| Darkroom (10x10) | $89\%/31.1K/1.7K$ | $92.3\%/15.7K/570$ | $91.7\%/2.8K/100$ | $75.6\%/20.8K/280$ |

$[16,128], n_a = 5$. The length of each sequence T is $12K$, resulting in a total of $6B$ time steps. For testing, we independently sample tasks $\mathcal{T}_{tst}$ with $n_s \in \{1, 16, 32, 64, 128\}$, ensuring each $n_s$ group contains 256 tasks.

The meta-training process is primarily conducted using 8 Nvidia Tesla A800 GPUs. We use a batch size of 5 per GPU, divided into segments (chunks) of $2K$ steps each. We optimize using the AdamW algorithm with a learning rate that decays from a peak value of $2 \times 10^{-4}$. The average time cost per iteration is 8 seconds for trajectories with $T = 12K$, and this cost increases linearly with sequence

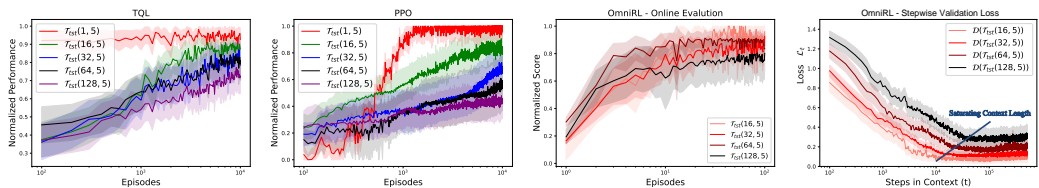

Figure 5: Left: Normalized episodic return of TQL-UCB, PPO and OmniRL on 64 AnyMDP evaluation tasks (mean $\pm 95\%$ CI). Hyper-parameters of PPO and TQL-UCB were separately tuned by grid-search inside every task family to maximize final-episode performance. Right: Per-step validation loss of OmniRL on a held-out static dataset, mirroring the online-RL trend.

length. For more details please check Appendix C.2. For the causal sequence model, we evaluate four architectures: RWKV-7 [70], Gated Delta-Net (GDN) [72], Gated Self-Attention (GSA) [67], Mamba2 [73]. The test results are largely consistent with the conclusions reported in language processing (RWKV-7 $\approx$ GDN > GSA > Mamba2, with details in Appendix D.1), demonstrating the capability of AnyMDP to serve as a benchmark for long-term sequence modeling. Therefore, we select RWKV-7 for subsequent experiments.

Without any further parameter tuning, we evaluate our model, namely *OmniRL*, on both unseen AnyMDP tasks in Figure 5, Gymnasium tasks, and DarkRoom [13] in Figure 16, and those performances are shown in Table 1. Notably, unlike previous ICRL works, our training set does not include any instances of DarkRoom. In our experiments, the selected tasks are constrained to environments with observation spaces of dimension $n_s \leq 128$ and action spaces of dimension $n_a \leq 5$. For environments with continuous observation spaces, such as *Pendulum*-v1, we manually discretize the observation space into 60 discrete classes using a grid-based discretization method. To adapt OmniRL that is trained with $n_a = 5$ to environments with less actions ($n_a < 5$), we reassign unused actions to valid ones. This further demonstrates the compatibility of OmniRL across environments with varying action space dimensions. We also found that proper reward shaping is important for OmniRL to work, as shown in Figure 13; the details can be found at Appendix C.3.

In Table 1, we compare the normalized performance, episode cost, and step costs of OmniRL, classical Tabular Q-learning (TQL) [60] with upper confidence bound (UCB) [61] (TQL-UCB for short), and Proximal Policy Optimization (PPO) [62]. OmniRL, meta-trained solely on AnyMDP, adapts effectively to most unseen tasks, confirming AnyMDP's representational power; by contrast, OmniRL trained on Garnet MDPs excels only on Garnet MDPs. The results also demonstrates OmniRL's superior sample efficiency, which aligns with prior ICRL findings. Notably, despite being trained solely on single-agent tasks, OmniRL adapts to multi-agent tasks like Switch2 by configuring observation spaces, enabling emergent inter-agent cooperation without explicit multi-agent interaction during training and thus decoupling cooperative behavior emergence from centralized mechanisms. Furthermore, in line with expectations, solving AnyMDP tasks becomes more difficult with increased state($n_s$) or action($n_a$) space size, with PPO proving more sensitive to action space extension and TQL-UCB more vulnerable to state space growth, as illustrated in Figure 9.

## 5.2 OmniRL Performs Both Offline and Online Learning Better

For the ablation study and comparison with the other methods including AD, AD$^\epsilon$, and DPT, we collect a smaller dataset with $|\mathcal{D}_{small}|$ comprising $128K$ sequences for training, where $n_s = 16, n_a = 5, T = 8K$, with a total of $1B$ time steps. Figure 6 summarizes the performance of different methods trained on $\mathcal{D}_{Small}$ with identical training iterations. The comparison includes AD, AD$^\varepsilon$, DPT, OmniRL, and OmniRL (w/o a priori) where the prior info $g_t$ is removed from the sequence.

We examine the performance of different methods with different initial contexts: (1) Online-RL: The agent starts with an empty trajectory ($h_0 = \emptyset$). (2) Offline-RL: The agent starts with an existing context derived from imperfect demonstrations (e.g., disturbed oracle policy) ($h_0 = h^\pi$). (3) Imitation Learning: The agent starts with an existing context derived from oracles($h_0 = h^{(exp)}$). For all three categories, the subsequent interactions are continually added to the context. Therefore, the models differ only in their initial memory or cache. The evaluation assesses the agents' abilities in two key areas: their capacity to exploit existing information and their ability to explore and exploit continually. In the results in Figure 6, OmniRL and OmniRL (w/o a priori) surpass AD, AD$^\varepsilon$, and DPT with large gap, validating the effectiveness of Step-wise Supervision (SS). OmniRL (w/o a priori) lags behind OmniRL with a noticeable gap in all three groups, demonstrating the effectiveness of integrating the prior information. Table 2 and Figure 17 further demonstrate that the offline-learning ability of OmniRL can generalize to Gymnasium environments.

## 5.3 Emergence of General-Purpose ICRL by Increasing Task Number

We validate task diversity's critical role in ICRL via independent meta-training across four datasets, each $\mathcal{D}(\mathcal{T}_{tra}(16, 5))$ containing $128K$ sequences but differing in task numbers ($|\mathcal{T}_{tra}| \in \{100, 1K, 10K, 128K\}$). Note that different trajectories can be generated from a single task, arising from the diverse behavior policies and random sampling in both decision and transition. We examine how the validation losses $\mathcal{L}_t$ on both seen and unseen tasks change with the number of meta-training

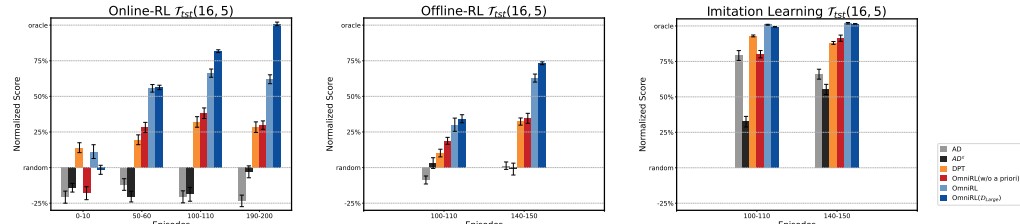

Figure 6: AD, AD$^\epsilon$, DPT, and OmniRL on 32 AnyMDP tasks, each tested under three initial contexts tailored for online RL, offline RL, and imitation learning. offline-RL and IL agents are seeded with 100 demonstration episodes before any interaction begins; results highlight the gains from DPD and prior-information integration.

Table 2: Offline-RL and IL performance of meta-trained OmniRL versus Conservative Q-Learning (CQL) [74] on four unseen Gymnasium tasks, using demonstrations from both oracle and random policies as initial contexts. Returns in episodes 0–20 measure offline RL and IL efficacy; returns in episodes 180–200 show the benefit of subsequent online RL.

| ENVIRONMENTS | TEACHER(PERFORMANCE) | 0∼20/180∼200 EPISODES PERFORMANCE | |
|---|---|---|---|
| | | **CQL** | **OMNIRL** |
| FROZENLAKE (SLIPPERY) | ORACLE (77.80%) | 76.61% / 76.11% | 70.12% / 77.27% |
| | RANDOM (1.46%) | 37.79% / 72.36% | 54.3% / 67.38% |
| CLIFF | ORACLE (-13) | -30.8 / -13 | -13 / -13 |
| | RANDOM (-109.84) | -560.2 / -16.2 | -91 / -17 |
| DISCRETE-PENDULUM (G=5) | ORACLE (-153.81) | -605.89 / -258.22 | -180 / -127 |
| | RANDOM (-941.65) | -1062.40 / -184.29 | -646 / -208 |
| DARKROOM (10X10) | ORACLE (0.22) | 0.23 / 0.22 | 0.22 / 0.21 |
| | RANDOM (-15.07) | -4.05 / 0.21 | 0.09 / 0.19 |

iterations (outer-loop steps) and steps in context $t$ (inner-loop steps) simultaneously; the results are shown in Figure 7. We remark the following observations:

**Task scale and diversity are crucial to the generalization of ICRL**. The previous investigation on ICL [25, 28] emphasizes the importance of "burstiness". Our results demonstrate for the first time that even when using "bursty" sequences alone, both the number of tasks and their overall diversity remain critically important. Specifically, in groups with $|\mathcal{T}_{\text{train}}| \leq 10K$, over-training leads to a transiency of ICL [75] in unseen tasks but a continued improvement in seen tasks. These findings confirm and extend the discovery of the "task identification" phase mentioned in Kirsch et al. [34], Pan et al. [76]. Drawing on the theories of ICL and IWL in Chan et al. [77], a possible explanation is that IWL dominates performance, with the model memorizing tasks and ICL selecting the correct one, leading to fast seen-task adaptation but poor unseen generalization. As the number of tasks increases continuously, the model becomes more dependent on ICL since memorizing task-specific information becomes less feasible. This is characterized by the improved generalization to unseen tasks and longer adaptation periods in both seen and unseen tasks, as shown in Figure 7.

**Long-context adaptation is a tax to pay for generalization scope**. Our results highlight a key insight on ICRL evaluation. Most previous ICRL works assess performance based on the average results over a fixed, short context span. However, our findings indicate that more generalized in-context learners may actually perform worse in zero-shot and even few-shot evaluations, particularly when there is significant overlap between the training and evaluation sets, i.e., when evaluation sets are closer to seen tasks. Therefore, we argue that it is more critical to focus on the *asymptotic performance* of a learner. This can be effectively evaluated by examining the performance at the final steps or episodes of a sufficiently long context, rather than short-term metrics.

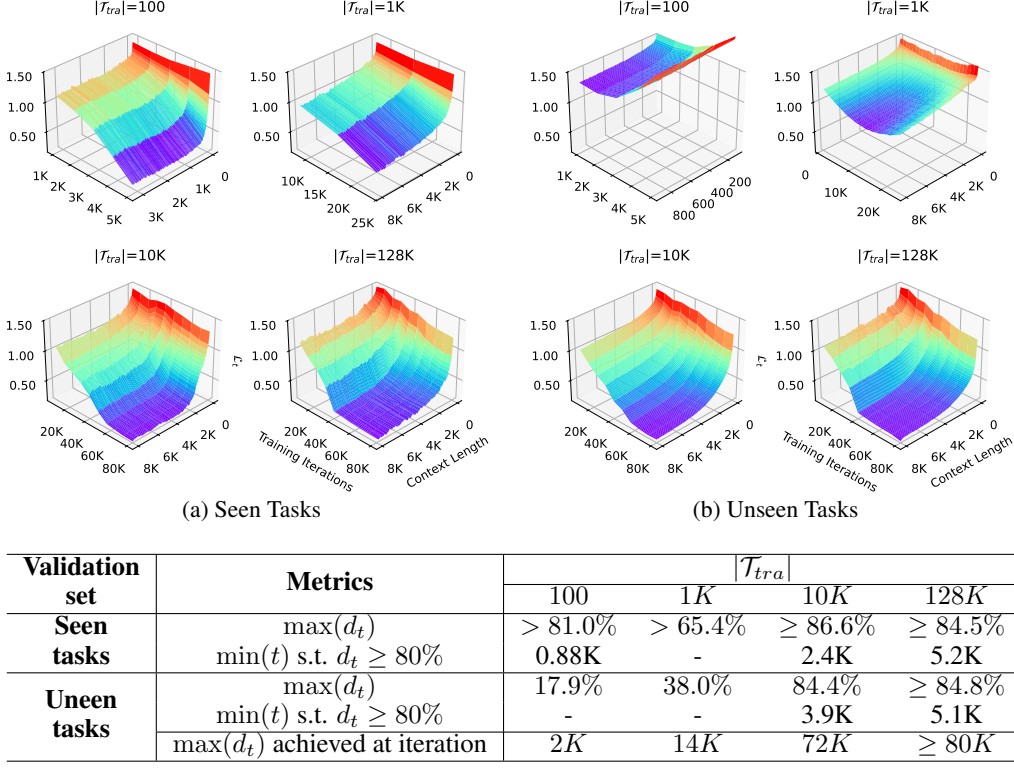

(a) Seen Tasks         (b) Unseen Tasks

| Validation set | Metrics | $|\mathcal{T}_{tra}|$ | | | |
|---|---|---|---|---|---|
| | | 100 | $1K$ | $10K$ | $128K$ |
| Seen tasks | $\max(d_t)$ | $> 81.0\%$ | $> 65.4\%$ | $\geq 86.6\%$ | $\geq 84.5\%$ |
| | $\min(t)$ s.t. $d_t \geq 80\%$ | 0.88K | - | 2.4K | 5.2K |
| Uneen tasks | $\max(d_t)$ | $17.9\%$ | $38.0\%$ | $84.4\%$ | $\geq 84.8\%$ |
| | $\min(t)$ s.t. $d_t \geq 80\%$ | - | - | 3.9K | 5.1K |
| | $\max(d_t)$ achieved at iteration | $2K$ | $14K$ | $72K$ | $\geq 80K$ |

Figure 7: Position-wise validation losses ($\mathcal{L}_t$, where lower values indicate better performance) and their properties on both seen and unseen tasks across meta-training iterations, varying context lengths, and variant number of tasks $|\mathcal{T}_{tra}|$. Each of the 4 groups of training data had $128K$ sequences, which were generated from $100, 1K, 10K,$ and $128K$ tasks, respectively. Each dataset underwent meta-training for up to $80K$ iterations. In the table, the notation ">" indicates values that could not be fully determined due to training being stopped early when performance on unseen tasks began to deteriorate. The normalized gain of ICL is defined as $d_t = 1 - \mathcal{L}_t/\mathcal{L}_0$, representing the improvement of performance as the context ($t$) increases. The table summarizes key findings of this study: as the number of tasks increases, the minimum step cost required to achieve an $80\%$ normalized gain of ICL also increases.

## 6 Conclusions and Discussions

We introduce AnyMDP, a scalable, low-bias task suite for benchmarking In-Context Reinforcement Learning (ICRL), and a companion framework featuring two innovations: Decoupled Policy Distillation and prior-information induction. Trained solely on AnyMDP, our model OmniRL outperforms prior methods in generalization, mastering wider task families and supporting more diverse learning paradigms.

**Broader impact**: Complementing prior studies, our findings highlight that task diversity and long sequence length are key determinants of general-purpose ICRL. Our results also indicate that long context is a necessary cost for generalization, thus advocate shifting evaluation metrics toward asymptotic performance measures. This work further motivates the construction of carefully curated synthetic datasets specifically designed for large-scale meta-training.

**Limitations and future work**: While procedural Discrete MDPs provide clean benchmarks for ICRL and long-context modeling, their extension to continuous state–action spaces remains bottlenecked. We recognize that a central challenge in this direction is to balance task complexity, task diversity, and data integrity while simultaneously calibrating generalization scope against practical constraints.

## Acknowledgments and Disclosure of Funding

This project is supported by Longgang District Shenzhen's "Ten Action Plan" for supporting innovation projects (Grant LGKCSDPT2024002), Guangdong Provincial Leading Talent Program (Grant No. 2024TX08Z319), National Natural Science Foundation of China (62033012), Guangdong S&T Program (Grant No.2025B0909040003), and the "Zhiguo" Action of Guangxi Science and Technology Program under Grant No.ZG2503980003.

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

# A    Notations

# B    Additional Information on AnyMDP

## B.1    Proof of Theorem 1

We began by establishing the upper bound for $p^{\mathfrak{r}}(s_j)$. Notably, if $\mathcal{S}_E \neq \emptyset$ the upper bound would be further reduced for indices $j > b$, as system termination resets states to indices $j < b_0$, Consequently, it suffices to analyze the case where $\mathcal{S}_E = \emptyset$ as non-empty $\mathcal{S}_E$ only tightens the bound.

Under the conditions that $\sum_{j<i} P_{\mathfrak{r}}(s_j|s_i) > \eta$ if $i > b_-$ and $\forall j > i + b_+$, $P_{\mathfrak{r}}(s_j|s_i) = 0$, we construct a **worst case transition kernel**. This kernel is defined as follows:

$\forall s_i$ with $n_s - b_+ > i > b$, $P_+(s_{i-1}) \equiv \eta$, $P_+(s_{i+b_+}|s_i) \equiv 1 - \eta$, $P_+(\cdot|s_i) \equiv 0$ for other cases.

It follows directly from this construction that $p_{\mathfrak{r}}(s_j) < p_+(s_j)$ for $j > b$, as $P_+$ maximizes transition probabilities to later states under the given constraints.

By the definition of the SD, we have:

$$\eta p_+(s_{i-1}) + (1-\eta)p_+(s_{i+b_+}) = p_+(s_i), \tag{6}$$

where $p_{+,i}$ is the SD of $s_i$ under transition $P_+$. Then, we define that $\delta = 1 - p_+(s_i)/p_+(s_{i-1})$, from which we derive the following:

$$p_+(s_{i+b_+})/p_+(s_i) = \frac{1-\delta-\eta}{(1-\eta)(1-\delta)}. \tag{7}$$

By applying the conditions $0 < \delta < 1/(b_+ + 1)$ and $\eta > 1.0 - \delta$, we can further bound the expression in Equation (7) as follows:

$$p_+(s_{i+b_+})/p_+(s_i) < (1-\delta)^{b_+}. \tag{8}$$

Equation (8) proves that $p_+(s_i)$ decays at a rate faster than $(1-\delta)$ as $i$ increases. Consequently, the SD $p_{\mathfrak{r}}$ also decays faster than $(1-\delta)$.

To establish **the lower bound**, we can utilize the constraint $\sum_{j>i} P_{\mathfrak{r}}(s_j|s_i) > \epsilon$ and construct a worst-case transition matrix as follows:

$\forall s_i$ with $n_s - 1 > i > b$, $P_-(s_{i-b_-}) \equiv 1 - \epsilon$, $P_-(s_{i+1}|s_i) \equiv \epsilon$, $P_-(\cdot|s_i) \equiv 0$ for other cases.

By analyzing this construction, we can validate that $p_-(s_i)$ decays at a rate slower than $\epsilon$ when $i$ increases. Consequently, the SD $p_{\mathfrak{r}}$ also decays slower than $\epsilon$. This completes the proof of Theorem 1.

Table 3: Default simplifications and notations used throughout the paper.

| | |
|---|---|
| ICL | IN-CONTEXT LEARNING |
| IWL | IN-WEIGHT LEARNING |
| ICRL | IN-CONTEXT REINFORCEMENT LEARNING |
| MC | MARKOV CHAIN |
| MDP | MARKOV DECISION PROCESS |
| SS | STEP-WISE SUPERVISION |
| SD | STATIONARY DISTRIBUTION |

| | |
|---|---|
| $\mathcal{S}$ | STATE OR OBSERVATION SPACE |
| $\mathcal{S}_0$ | STATES FOR RESET |
| $\mathcal{S}_E$ | STATES TRIGGERING TERMINATION |
| $T_E$ | MAXIMUM LENGTH OF AN EPISODE |
| $\mathcal{P}(s, a, s')$ | TRANSITION FUNCTION OF $s, a \rightarrow s'$ |
| $\mathcal{R}(s, a, s')$ | REWARD FUNCTION OF $s, a \rightarrow s'$ |
| $\gamma$ | DISCOUNT FACTOR FOR REWARDS |
| $\mathcal{A}$ | ACTION SPACE |
| $\Pi$ | A SET / COLLECTION OF POLICIES |
| $s \in \mathcal{S}$ | STATE OR OBSERVATION |
| $a \in \mathcal{A}$ | ACTION |
| $r \in \mathbb{R}$ | REWARD OR FEEDBACK |
| $g$ | PRIOR INFORMATION OF ACTION |
| $\pi \in \Pi$ | POLICY FUNCTION |
| $Q^{\pi}(s, a)$ | STATE-ACTION VALUE FUNCTION |
| $V^{\pi}(s, a)$ | STATE VALUE FUNCTION, $V^{\pi}(s) = \mathbb{E}_{a \sim \pi}[Q^{\pi}(s, a)]$ |
| $\pi^*$ | ORACLE POLICY ACHIEVING HIGHEST EXPECTED EPISODIC REWARDS |
| $a^* \sim \pi^*$ | ACTION GENERATED BY ORACLE POLICY |
| $Q^*, V^*$ | VALUE FUNCTION WITH ORACLE POLICY |
| $\tau(n_s, n_a)$ | TASK WITH $n_s$ DISCRETE STATES AND $n_a$ DISCRETE ACTIONS |
| $\mathcal{T}(n_s, n_a)$ | A COLLECTION OF TASKS $\tau(n_s, n_a)$ |
| $\mathcal{D}(\mathcal{T})$ | DATASET RECORDED FROM EXECUTION ON $\mathcal{T}$ |
| $h_t$ | A TRAJECTORY $[(s_1, g_1, a_1, r_1), ..., (s_t, g_t, a_t, r_t)]$ |
| $l_t$ | STEP-WISE SUPERVISION LABELS FOR TRAJECTORY $h_t$ |
| $p_{\mathfrak{r}}(s)$ | STATIONARY DISTRIBUTION (SD) OF MDP WITH THE POLICY $\pi^{\mathfrak{r}}$ |
| $\theta$ | PARAMETERS OF A MODEL, KEPT UNCHANGED IN THE INNER LOOP (ICL) |
| $\phi$ | CACHES OR MEMORIES, STARTS FROM SCRATCH IN THE INNER LOOP |
| $\mathfrak{o}$ | AGENT WITH ORACLE POLICY $\pi^*$ |
| $\mathfrak{q}$ | Q-LEARNING AGENT |
| $\mathfrak{r}$ | AGENT WITH RANDOM POLICY |
| $\mathfrak{m}$ | MODEL-BASED REINFORCEMENT LEARNING AGENT |
| $\mathfrak{o}^{\varepsilon}$ | AGENT $\mathfrak{o}$ DISTURBED WITH A DECAYING NOISE $\varepsilon$ |

## B.2 Details of Procedural Generation

Algorithm 1 elaborates on the detailed procedural generation of AnyMDP tasks.

**Transition Sampling**. The generation process commences by randomly sampling a ranking of states, denoted as $s_1, ..., s_{n_s}$. The initial state set $\mathcal{S}_0$ are sampled from $\{s_j | j < n_0\}$, while the absorbing state set (including pitfalls and goals) $\mathcal{S}_E$ is randomly sampled. The goals are constrained only to $s_{n_s}$. Subsequently, the average transition matrix is sampled, and based on this matrix, it is randomly decomposed into a state-action transition matrix. Notice that the state-action transition is constructed utilizing the average state transition kernel and the weights $w_i(a_k)$, as follows:

$$\mathcal{P}(s_j | s_i, a_k) = P_{\mathfrak{r}}(s_j | s_i) \frac{\exp\left[-(w_i(a_k) - j)^2 / \sigma_k^2\right]}{\sum_l \exp\left[-(w_i(a_l) - j)^2 / \sigma_l^2\right]} \tag{9}$$

Ergodicity in the average transition of AnyMDP is checked by computing the stationary distribution for all initial states and evaluating the variance among them. In practice, we observed that the majority of sampled transition matrices adhering to banded transition constraints exhibit ergodicity, with non-ergodic scenarios being exceedingly uncommon.

While the theoretical bound in Theorem 1 provides valuable non-triviality guarantees, it remains inherently conservative due to the analytical challenges posed by randomly generated task structures.

---

**Algorithm 1** AnyMDP $TaskSampler$

---

1: **Input:** $n_s, n_a$ **Returns:** $\tau(n_s, n_a)$
2: Randomly generate an ordered list arranged from low-valued to high-valued states $S = [s_1, s_2, ..., s_{n_s}]$ by permute $\mathcal{S} = \{1, ..., n_s\}$
3: Randomly sample states for reset $\mathcal{S}_0 \subset \{s_0, s_1, ..., s_{b_0}\}$
4: Randomly sample states triggering termination $\mathcal{S}_E \subset \mathcal{S}/\mathcal{S}_0$
    ▷ Sample banded state transition matrix under uniform random policy
5: **for** $i \leftarrow 1$ to $n_s$ **do**
6:     Randomly sample $\hat{P}_{\mathfrak{r}}(\cdot|s_i)$, s.t. banded transition constraints (Equation (1))
7: **end for**
    ▷ Ensure the state transition has a stationary distribution
8: Calculate $\bar{P}_{\mathfrak{r}} = (P_{\mathfrak{r}})^K$ by considering $\mathcal{S}_0, \mathcal{S}_E$; if $\mathbb{E}_j[\mathbb{V}_i(\bar{P}_{\mathfrak{r}}[s_j|s_i])] > \varepsilon$, go back to resample $P_{\mathfrak{r}}$.

    ▷ Sample state-action transition with the constraint of the state transition
9: **for** $i \leftarrow 1$ to $n_s$ **do**
10:     $\forall k \in [1, n_a]$, randomly sample scalars $w_i(a_k) \in [i - b_-, i + b_+]$, $\sigma_k \sim Exp(1)$
11:     Sample state-action transition $\mathcal{P}(s_j|s_i, a_k)$ by applying Equation (9)
12: **end for**
    ▷ Sample composite reward function and ensure the value function distribution is as expected
13: Randomly sample state-action-dependent rewards $r_{sa} \in \mathbb{R}^{n_s \times n_a}$
14: Randomly sample state-dependent potential $v_s \in \mathbb{R}^{n_s}$
15: **repeat**
16:     Randomly sample state-dependent reward function $r_s \in \mathbb{R}^{n_s}$, s.t. $\forall j > i, r_s(s_j) \geq r_s(s_i)$
17:     Set $\mu_{\mathcal{R}}(s, a, s')$ by composite rewards (Equation (10))
18:     Calculate value function $V^*(s)$ based on $\mathcal{P}, \mathcal{R}, \mathcal{S}_0, \mathcal{S}_E$
19: **until** Ascending value function condition is satisfied (Equation (2))
    ▷ Ensure the optimal trajectory is not trivial
20: **Final validation:** calculate oracle policy steady state distribution ($p_{\mathfrak{o}}$)
21:         if $-\sum_s p_{\mathfrak{o}} \log p_{\mathfrak{o}} / \log n_s > H_0$ then accept, else resample the task.

---

Empirically, we observe that setting $b_+ \leq n_s/4$, $b_- \geq n_s/2$, $\epsilon > 1.0e - 3$, $\eta > 0.5$ is enough to consistently yield high-quality Markov chain formulations.

**Composite Reward Sampling**. To further enhance the quality of the task, rather than naively sampling the reward function $\mathcal{R}$ from $\mathbb{R}^{n_s \times n_a \times n_s}$, e independently sample the mean reward matrix $\mu^{\mathbf{R}}$ and the noises $\Sigma^R$. This ensures that the reward $r(s, a, s') \sim \mathcal{N}(\mu^R_{s,a,s'}, \Sigma^R_{s,a,s'})$. The mean reward matrix $\mu_{R,s,a,s'}$ is sampled from *composite reward functions*, which is defined as follows:

$$\mu^R_{s,a,s'} = r^s(s') + r^{sa}(s, a) + v^s(s) - v^s(s') \tag{10}$$

We first sample the state-dependent reward $r^s$ based on the ranking of states $s_1, ..., s_{n_s}$ and whether each state corresponds to pitfalls or goals. Subsequently, we sample the disturbance reward function $r^{sa}$ and state-value function $v^s$, which can be interpreted as random state-action costs and random reward shaping terms, respectively.

To ensure non-trivial task formulations, we implement a suite of validation checks, including: 1. Ergodicity check on the average transition function; 2. Ascending value function check to guarantee monotonic improvement; 3. Minimum threshold check on the normalized entropy $\mathcal{H}(p_{\mathfrak{o}})$ of the state distribution (SD) under the oracle policy $\mathfrak{o}$. If any of these checks fail, we either resample the transition dynamics or readjust the sampling of composite rewards. Details of the sampling process are provided in the Appendices.

## B.3 Additional Properties of AnyMDP

**Computational Efficiency of Task Sampling**. Figure 8 illustrates the computational cost of generating AnyMDP tasks for various state space sizes $n_s \in \{8, 16, 32, 64, 128\}$. Notably, $n_s = 8$ exhibits significantly higher computation times. This is primarily due to the frequent resampling required when final validation check fails. It is important to note that the AnyMDP task generation process was executed on single CPU. Given this, the use of readily available parallelization techniques could significantly accelerate task generation.

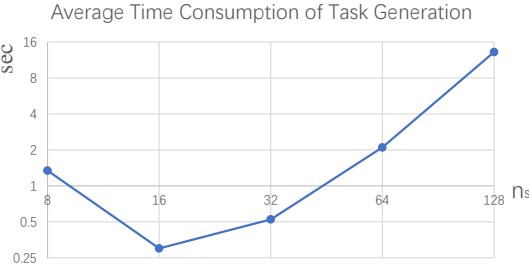

Figure 8: Time consumption of AnyMDP task generation on an a Intel(R) Xeon(R) Platinum 8374C CPU.

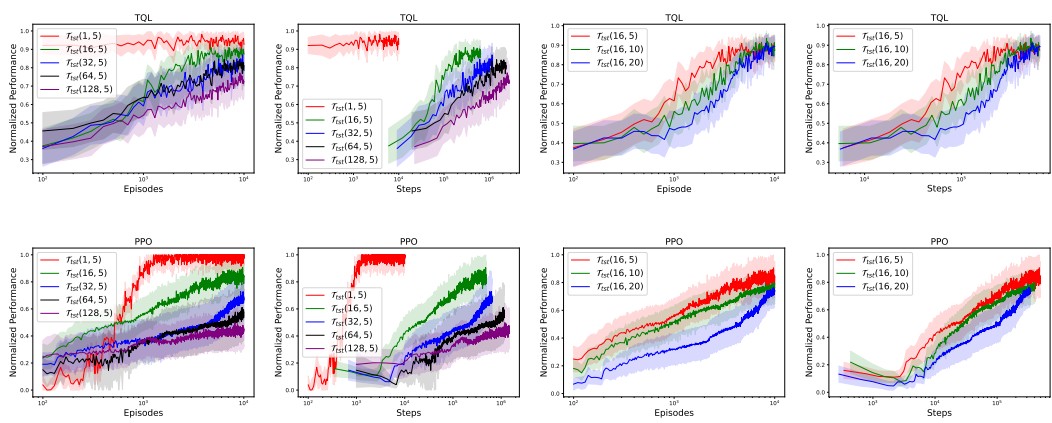

Figure 9: Performance of Tabular Q-Learning and PPO on AnyMDP tasks of variant state space and action spaces, with respect to episodes and steps.

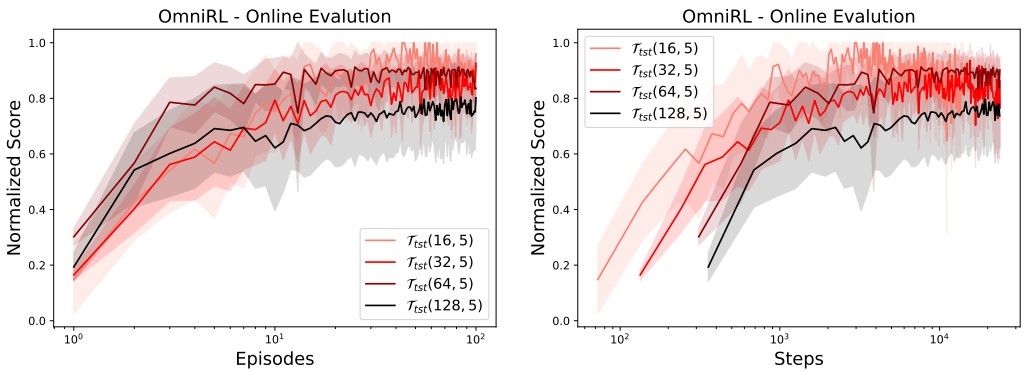

Figure 10: Performance of OmniRL on AnyMDP tasks of variant state space, with respect to steps.

**Relations between solution difficulty and the state–action space**. Figure 9 illustrates the performance of Tabular Q-Learning and Proximal Policy Optimization (PPO) on AnyMDP tasks with varying state space and action space sizes. The results indicate that increasing either the state space size ($n_s$) or the action space size ($n_a$) enhances the complexity of the task, as evidenced by the need for more training steps to achieve convergence. OmniRL's result on AnyMDP tasks with different state spaces also supports this phenomenon, shown in Figure 10. Additionally, an increase in the state space size ($n_s$) leads to a higher number of steps per episode.

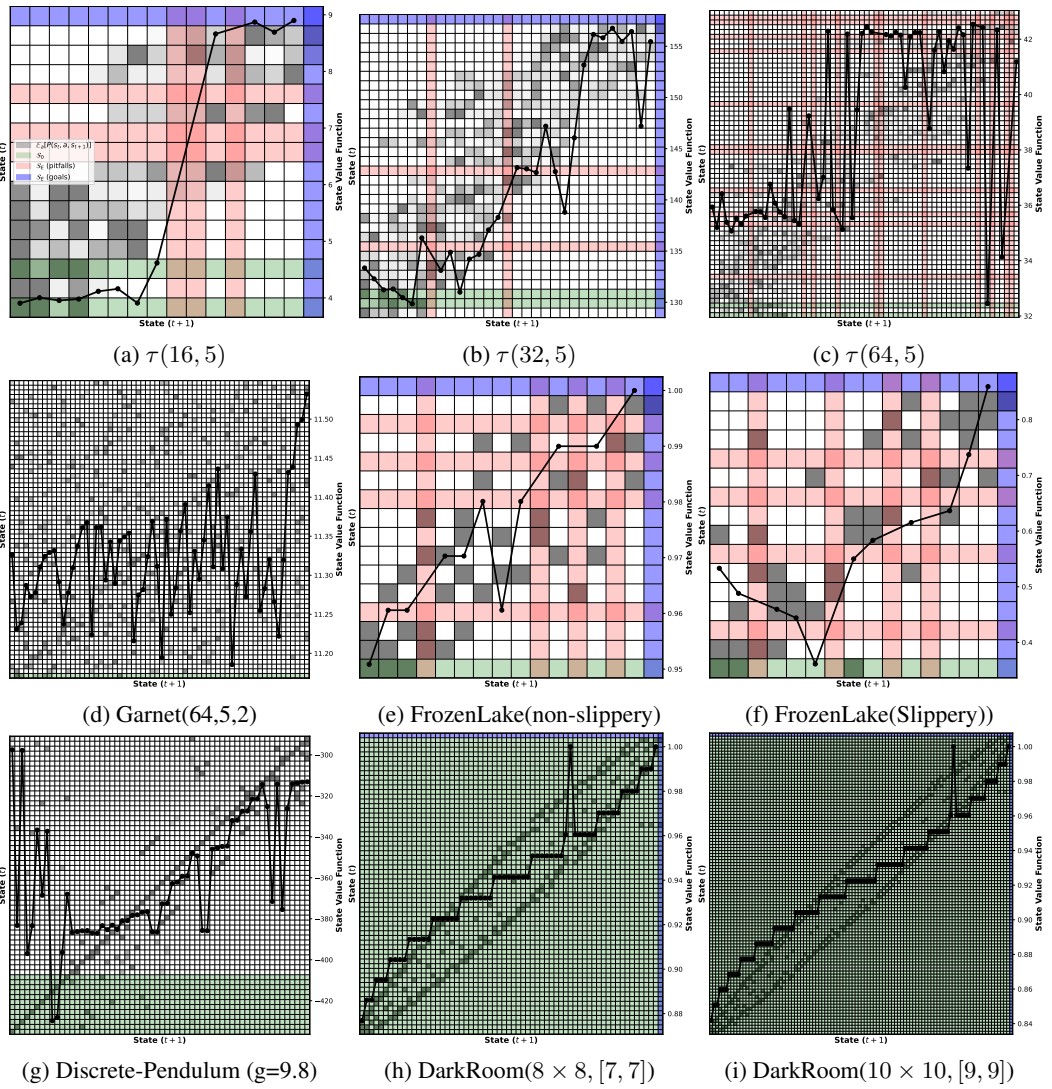

Figure 11: Visualization of three tasks sampled from AnyMDP, with the number of states $n_s$ varying across $\{16, 32, 64\}$, Garnet$(64, 5, 2)$, and Gymnasium tasks including non-slippery and slippery *FrozenLake* and *Discrete-Pendulum*, and 2 tasks sampled from DarkRoom. States are reordered according to the SD ($p_\mathbf{r}$), ordered from high to low. Gray blocks indicate positive state transition kernels. Red and blue blocks mark pitfalls ($\mathcal{S}_E^-$) and goals ($\mathcal{S}_E^+$), respectively, which trigger episode termination. Green blocks mark $\mathcal{S}_0$. The black line denotes the state value function under the optimal policy $V^*(s)$. Notably, AnyMDP is capable of generating a diverse range of tasks, including those with and without pitfalls and goals. The visualizations demonstrate that tasks generated by AnyMDP can be of comparable quality to those designed by humans. A common principle observed is that higher rewards are often associated with more challenging goals, akin to the concept of "high-hanging fruit".

Table 4: Summarizing the data synthesis strategies of different methods under the Decoupled Policy Distillation (DPD) framework.

| DATA SYNTHESIS PIPELINE | BEHAVIOR POLICIES ($\Pi$) | REFERENCE POLICY |
|---|---|---|
| AD [13] | $\mathfrak{q}$ | $\mathfrak{q}$ |
| AD$^\epsilon$ [15] | $\mathfrak{o}^\varepsilon$ | $\mathfrak{o}^\varepsilon$ |
| DPT [16] | $\mathfrak{o}, \mathfrak{q}, \mathfrak{r}$ | $\mathfrak{o}$ |
| DECOUPLED POLICY DISTILLATION (OURS) | $\mathfrak{o}, \mathfrak{q}, \mathfrak{m}, \mathfrak{r}, \mathfrak{o}^\varepsilon$ | $\mathfrak{o}$ |

**Visualizing discrete MDPs**. Following the previous analysis, for any discrete Markov Decision Process (MDP), we can rearrange the states such that the SD $p_\mathfrak{r}$ decreases monotonically. We then plot the transition kernel $P_\mathfrak{r}$ in this rearranged order. In the visualization, we use varying opacity to represent the elements of $P_\mathfrak{r}$ and different colors to distinguish the initial states $\mathcal{S}_0$, positively rewarded terminal states (goals) $\mathcal{S}_E^+$, and negatively rewarded terminal states $\mathcal{S}_E^-$. This visualization, shown in Figure 11, enables us to analyze both procedurally generated AnyMDP tasks and human-designed Gymnasium tasks. Several interesting observations can be made:

- **Higher rewards for higher effort**. Both procedurally generated AnyMDP tasks and human-designed Gymnasium tasks exhibit a negative correlation between the SD $p_\mathfrak{r}$ and the value function $V^*$. This suggests a common principle: states with lower SD probability tend to have higher value functions, akin to the concept of "high hanging fruit".

- **Banded transition kernel**. When ordered by decreasing SD probability, the transition kernels of all Markov chains display the characteristics of a banded matrix. This observation further validates the effectiveness of the procedural generation method outlined in Algorithm 1.

## C Additional Information on Experiment Settings

### C.1 Data Synthesis

---
**Algorithm 2** Data Synthesis Pipeline
---
**Input:** $\mathcal{T}$, $N_{sample}$, Collection of behavior policies $\Pi$, reference policy $\pi^*$
**set:** $\mathcal{D}(\mathcal{T}) = \emptyset$
**for** $[1, N_{sample}]$ **do**
  **sample:** task $\tau \sim \mathcal{T}$
  **set:** $t = 0, h_0 = [], l_0 = []$
  **repeat**
    **sample:** behavior policy $\pi^{(b)} \sim \Pi$
    **reset:** $\tau$ and update $s_t$
    **repeat**
      **sample:** $a_t \sim \pi^{(b)}(a|s)$, $g_t = tag(\pi^{(b)})$
      **sample:** $a_t^* \sim \pi^*(a|s)$
      **execute:** $a_t$ in $\tau$ and obtain $s_{t+1}$, $r_t$
      **set:** $h_t = h_{t-1} \oplus [s_t, g_t, a_t, r_t], l_t = l_{t-1} \oplus a_t^*, t = t + 1$
    **until** Episode is over
  **until** $t \geq T$
  **Set:** $\mathcal{D}(\mathcal{T}) = \mathcal{D}(\mathcal{T}) \cup \{h_T, l_T\}$
**end for**
**Return:** $\mathcal{D}(\mathcal{T})$

---

The data synthesis pipeline of OmniRL involves generating diverse trajectories $h$ using a variety of behavior policies and creating step-wise labels $l$ with an oracle policy $\pi^\mathfrak{o}$. This pipeline is detailed in Algorithm 2. We incorporate at least five distinct types of agents:

- An agent with the oracle policy ($\mathfrak{o}$),

Table 5: Correpsondance of prompt IDs and the policies it represents.

| ID $(g)$ | AGENT TYPE | DESCRIPTION |
|---|---|---|
| 0 | $\mathfrak{o}(\gamma = 0)$ | SINGLE-STEP GREEDY |
| 1 | $\mathfrak{o}(\gamma = 0.5)$ | MYOPIC GREEDY |
| 2 | $\mathfrak{o}(\gamma = 0.93)$ | SHORT-TERM ORACLE |
| 3 | $\mathfrak{o}$ | ORACLE WITH $\gamma > 0.99$ |
| 4 | $\mathfrak{m}$ | MODEL-BASED REINFORCEMENT LEARNER |
| 5 | $\mathfrak{q}$ | TABULAR Q LEARNER |
| 6 | $\mathfrak{r}$ | RANDOMIZED POLICY (INCLUDING PERTURBED ACTION IN $\mathfrak{o}^\varepsilon$) |
| 7 | UNK | RESERVED ID |

- An agent with a randomized policy ($\mathfrak{r}$),

- A tabular Q-Learning agent ($\mathfrak{q}$),

- A model-based reinforcement learning agent ($\mathfrak{m}$),

- An agent with the oracle policy perturbed by a decaying noise $\varepsilon$ ($\mathfrak{o}^\varepsilon$).

With these notations, Table 4 can be used to represent not only the data synthesis pipeline of OmniRL but also the previous imitation meta-training-based ICRL methods, including AD, AD$^\varepsilon$, and DPT, as shown in Table 4. Notably, the synthesis pipeline of OmniRL is most similar to that of DPT. However, there are key differences: OmniRL employs a more diverse set of behavior policies and incorporates step-wise supervision (SS).

For the prior information $g$, we assign eight different IDs to the actions with $g \in [0, 7]$ which originated from 8 types of different agents, as shown in Table 5. Specifically, we exclude the actions generated by the seven types of different agents and reserve $g = 7$. This reserved ID is used to replace the action ID approximately $15\%$ of the time steps with the data synthesis pipeline of Algorithm 2.

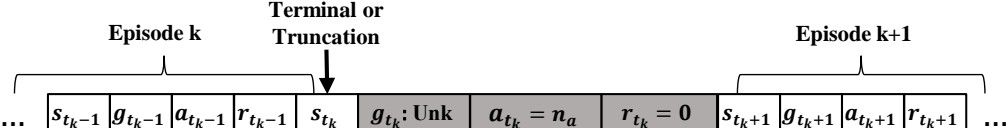

Figure 12: A sketch of the tokens used to denote the intervention between two episodes. An extra action token is introduced, which is distinct from the normal actions ($a_{t_k} = |\mathcal{A}| = n_a$).

**Addressing terminal states**: The presence of terminal and truncation states necessitates special handling in reinforcement learning. In OmniRL, we avoid explicitly adding a terminal or truncation token to the sequence. Instead, we encode terminal and truncation states by introducing an additional action $a$, which is maintained as distinct from the standard action space $\mathcal{A}$. Additionally, we assign a reward of 0 and set the prior information $p = 7$ for these special steps, as depicted in Figure 12.

## C.2 Meta-Training Details

**Model structures**: Before injection into causal models, the states ($s_t$) and actions ($a_t$) are encoded using embedding layers with a hidden size of $512$. The rewards ($r_t$) are treated as continuous features encoded by $1 \times 512$ linear layer. The sequence model has a hidden size of $512$, inner hidden size of $1024$, hidden ratio of $2$, and block number of $18$ for RWKV-7. The model has approximately $43.6$M total parameters ($42.9$M in RWKV-7 blocks), as shown in Table 6. We employ the open-source implementation of *flash-linear-attention* ♠.

**Meta-training**: Algorithm 3 outlines the detailed process of the meta-training procedure. Notably, we perform the backward pass segment-wise and accumulate the gradients. The gradients are not applied until the end of a sequence. We utilize a constant segment length $T_{k+1} - T_k = 2K$, which results in six backward passes for $T = 12K$ before applying the gradient.

---

♠https://github.com/fla-org/flash-linear-attention

Table 6: The parameter settings of the sequence model. Note that for different models, the relationship between head dimension, head number, and hidden size varies. We follow the settings used in *flash-linear-attention*.

|  | GDN | GSA | Mamba2 | RWKV7 |
|---|---|---|---|---|
| Block nums | 18 | 18 | 18 | 18 |
| Hidden size | 512 | 512 | 512 | 512 |
| Inner hidden size | $2 \times 512$ | $2 \times 512$ | $2 \times 512$ | $2 \times 512$ |
| Head nums | 8 | 8 | 8 | 8 |
| Head dim | 48 | 64 | 128 | 64 |
| Parameters | 46.3M | 42.9M | 31.6M | 42.9M |

---

**Algorithm 3** Meta-Training Process

---

**Input:** $\mathcal{D}(\mathcal{T}_{tra}), \mathcal{D}(\mathcal{T}_{tst})$
**for** epochs from 1 to maximum epochs **do**
    **for** $h_T, l_T \in \mathcal{D}(\mathcal{T}_{tra})$ **do**
        **set:** segments $K = T/T_{seg}$, gradients $g = \mathbf{0}$, initial memory $\phi_0 = 0$
        **for** $k \in [0, K)$ **do**
            **forward:** update $\phi_{k-1} \to \phi_k$ based on Equation (4), $\phi_{k-1}$, $h_{T_k:T_{k+1}}$ and $l_{T_k:T_{k+1}}$
            **backward:** calculating $g_k = \nabla \sum_{t \in [T_k, T_{k+1}]} w_t \mathcal{L}_t$ by stopping gradient of $\phi_{k-1}$
            **accumulate gradient:** $g = g + g_k$
        **end for**
        **apply gradient:** $g$ to update $\theta$
    **end for**
    **validate:** averaging $\mathcal{L}_t$ and $\mathcal{L}$ on $\mathcal{D}(\mathcal{T}_{tst})$
**end for**

---

Table 7 provides an overview of the primary datasets used in this study. For the $\mathcal{D}_{\text{Large}}$ dataset, the state space size $n_s$ is uniformly sampled from the range $[16, 128]$ to ensure robustness across varying state spaces. To evaluate the extrapolation capability of the model trained on $\mathcal{D}_{\text{Large}}$, we conducted a validation test with a context length of 1 million steps and observed that the loss began to gradually increase beyond $80K$ steps. Building upon this observation, we incorporated a post-training stage for long sequences with a context length of $512K$, the dataset is denoted as $\mathcal{D}_{\text{Long}}$.

---

**Algorithm 4** Evaluation Process

---

**Input:** $\mathcal{T}_{tst}$, collection of demonstration trajectories $\mathcal{H}_0 = \{h_0\}$,
**set:** $S^{eval} = \emptyset$
**for** $\tau \in \mathcal{T}_{tst}$ **do**
    **set:** $R_{max}$=average episodic reward of $\mathfrak{o}$, $R_{min}$=average episodic reward of $\mathfrak{r}$
    **set:** $\mathcal{S}^{eval}_\tau = []$
    **repeat**
        **retrieving:** $h_0$ from $\mathcal{H}_0$ according to $\tau$
        **reset:** $\tau$ and obtain $s_1$, $R = 0$
        **repeat**
            **sample:** $a_t \sim p_t^\theta$ with Equation (4)
            **execute:** $a_t$ in $\tau$ and obtain $s_{t+1}$, $r_t$
            **set:** $h_t = h_{t-1} \oplus [s_t, g_t, a_t, r_t]$ with $g_t =$"Unk"
            **set:** $R \leftarrow R + r_t, t \leftarrow t + 1$
        **until** Episode is over
        **calculate:** normalized performance $S^{eval}_\tau \leftarrow S^{eval}_\tau \oplus [\frac{R - R_{min}}{R_{max} - R_{min}}]$
        **set:** $N_{episodes} \leftarrow N_{episodes} + 1$
    **until** $N_{episodes} > N_{max}$
    **Record:** $S^{eval} \leftarrow S^{eval} \cup S^{eval}_\tau$
**end for**
**Return:** $S^{eval}$

---

Table 7: Details of the meta-training dataset

| DATASET | DESCRIPTION | TIME STEPS |
|---|---|---|
| $\mathcal{D}_{Small}$ | $n_s = 16, n_a = 5$ 
 $|\mathcal{T}_{tra}| = |\mathcal{D}(\mathcal{T}_{tra})| = 128K, SequenceLength = 8K$ | $1B$ |
| $\mathcal{D}_{Large}$ | $n_s \in [16, 128], n_a = 5$ 
 $|\mathcal{T}_{tra}| = |\mathcal{D}(\mathcal{T}_{tra})| = 512K, SequenceLength = 12K$ | $6B$ |
| $\mathcal{D}_{Long}$ | $n_s \in [16, 128], n_a = 5$ 
 $|\mathcal{T}_{tra}| = |\mathcal{D}(\mathcal{T}_{tra})| = 12K, SequenceLength = 512K$ | $6B$ |

$$reward = \begin{cases} 1, & \text{if reach goal} \\ -1, & \text{if reach hole} \\ 0, & \text{otherwise} \end{cases}$$

(a) FrozenLake-v1(slippery)

$$reward = \begin{cases} 1, & \text{if reach goal} \\ -1, & \text{if reach hole} \\ -0.05, & \text{otherwise} \end{cases}$$

(b) FrozenLake-v1(not slippery)

$$reward = \begin{cases} 1, & \text{if reach goal} \\ -1, & \text{if reach cliff} \\ -0.03, & \text{otherwise} \end{cases}$$

(c) CliffWalking-v0

$$reward = \max\left( \frac{reward}{30} + 0.1, -0.1 \right)$$

(d) Pendulum-v1

$$agent\ reward = \begin{cases} 1, & \text{if reach goal} \\ 0.08, & \text{if distance to goal decrease} \\ -0.12, & \text{if distance to goal increase} \\ -0.04, & \text{if still} \\ 0, & \text{if finish} \end{cases}$$

$$shared\ reward = \sum_{i=1}^{2} agent\ reward_i$$

(e) Switch

Figure 13: Reward shaping

### C.3 Evaluation Details

As shown in Algorithm 4, since the episode length and baseline average episodic reward vary significantly across different tasks, we normalize the episodic reward using the oracle policy (o) and the uniform random policy (τ). This normalization represents the percentage of oracle performance achieved. For AnyMDP, the evaluation averages the performances over $64$ variant unseen tasks. For Gymnasium tasks, the evaluation is conducted by averaging the results over 3 runs on the same task.

By default, the normalized performance $S^{\text{eval}}$ is averaged across tasks with identical $N_{\text{episodes}}$. The deviation is estimated using the $95\%$ confidence interval of the mean.

For Tabular Q-Learning and PPO, we conduct 5 episodes of testing after every 100 episodes of training for each run. We evaluate performance based solely on these test episodes. In contrast, for ICRL, we do not differentiate between training and testing phases.

When performing online inference with OmniRL, we do not employ any additional exploration strategies. Instead, we maintain a softmax sampling temperature of $0.5$. For offline learning, where demonstrations from the teacher are encoded, we use the original prior information $g_t$ without

modification. In contrast, for online learning, where actions are generated by the agent itself, we set $g_t = 7$ (Unk).

We apply some reward shaping to Gymnasium tasks as shown in Figure 13. OmniRL supports $n_s \leq 128$ and $n_a \leq 5$. For environments with $n_a < 5$, we find directly setting $a = a \mod n_a$ is enough, which also demonstrates the generalizability of OmniRL to variant action spaces.

# D    Additional Empirical Results

## D.1    AnyMDP as a long-context benchmark for procedural memory

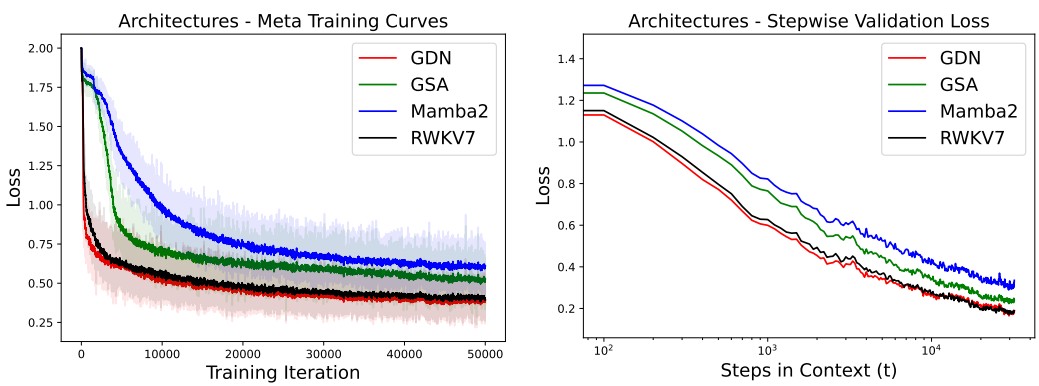

Figure 14: Comparison of meta-training dynamics across AnyMDP dataset of $6B$ time steps and step-wise loss (lower is better) on the validation set $\mathcal{D}(\mathcal{T}_{tst}(n_s \in [16, 128], n_a = 5))$ for different linear-attention models.

Figure Figure 14 reports the performance of different linear-attention models throughout AnyMDP training and on the held-out AnyMDP evaluation data ($\mathcal{T}(n_s \in [16, 128], n_a = 5)$), including Gated Delta Net (GDN), Gated Slot Attention (GSA), Mamba2, and RWKV-7. Error decreases polynomially with context length up to 20k tokens, yielding an almost straight line when plotted against the logarithm of context length. Consequently, AnyMDP serves as a long-context benchmark for procedural memory, whereas prior benchmarks such as Needle-in-the-Haystack primarily probe episodic memory.

## D.2    OmniRL achieves automatic trade-off between exploration and exploitation

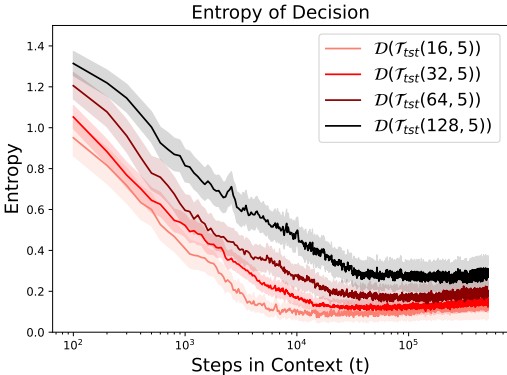

Figure 15: The position-wise entropy when validating RWKV-7 on different datasets.

Previous studies have noted that in-context reinforcement learning (ICRL) can automatically balance exploration and exploitation. This phenomenon has been theoretically linked to posterior sampling. In Figure 15, we illustrate the entropy of the decision-making process as a function of steps within the context. When compared to Section 5.1, we observe that the decrease in loss ($\mathcal{L}_t$) is primarily driven by the reduction in the entropy of the policy. Specifically, the agent initially assigns equal probabilities to all actions, reflecting an exploratory phase. As more contextual information accumulates, the agent gradually converges its choices, thereby transitioning towards exploitation. This empirical finding suggests that imitating an optimal policy (oracle) is sufficient to achieve an automatic balance between exploration and exploitation.

## D.3 Additional Evaluation on Gymnasium

Figure 16 and Figure 17 demonstrate OmniRL's online-RL, offline-RL, and imitation learning capabilities toward diverse unseen tasks.

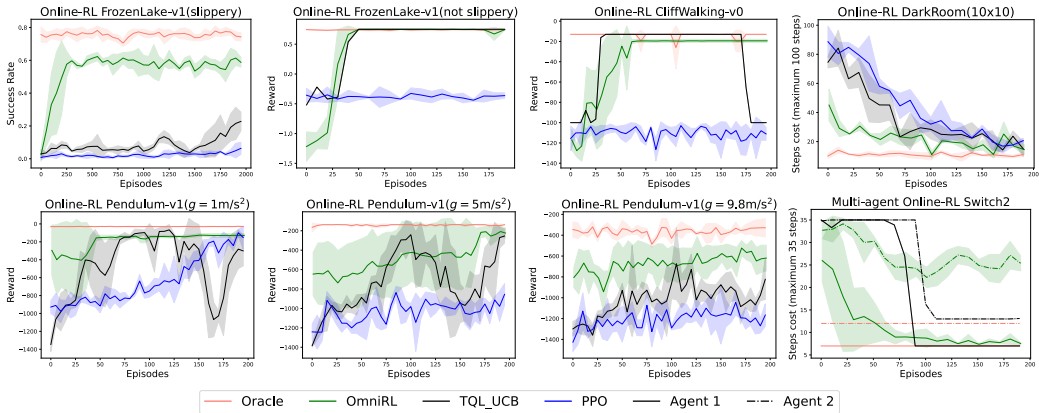

Figure 16: Selected online evaluation results for TQL-UCB, PPO, and OmniRL across Gymnasium environments. Notably, despite never having been exposed to these environments during training, OmniRL demonstrates strong adaptability by achieving competitive performance on most tasks with high sample efficiency.

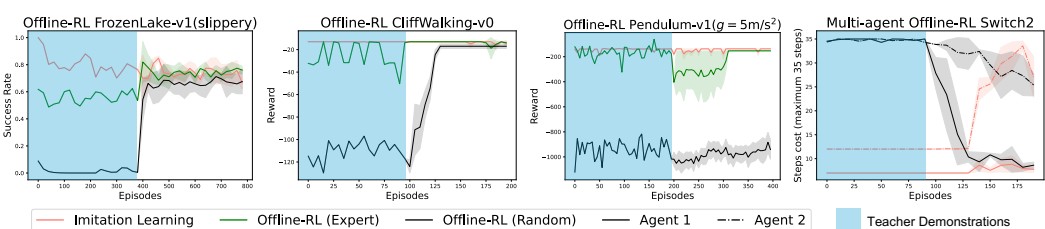

Figure 17: Selected offline evaluation results for OmniRL across Gymnasium environments, demonstrating the model's offline-RL and imitation learning capabilities toward unseen tasks.

## D.4 Memory states in ICRL implicitly encode the task structure

ICRL with linear attention captures all the information required to solve the environment in its memories ($\phi_t$). We perform a comprehensive t-SNE analysis to examine how these memories transform across different tasks during Online-RL evaluation. As shown in Figure 18, the clustering patterns confirm the distinct task distributions of Gym, Darkroom, and AnyMDP. Notably, Darkroom and Gym clusters are predominantly located in the top-left region, while AnyMDP occupies a broader spatial area, reflecting its greater diversity. This spatial differentiation emphasizes AnyMDP's unique characteristics and highlights OmniRL's strong generalization ability across diverse tasks.

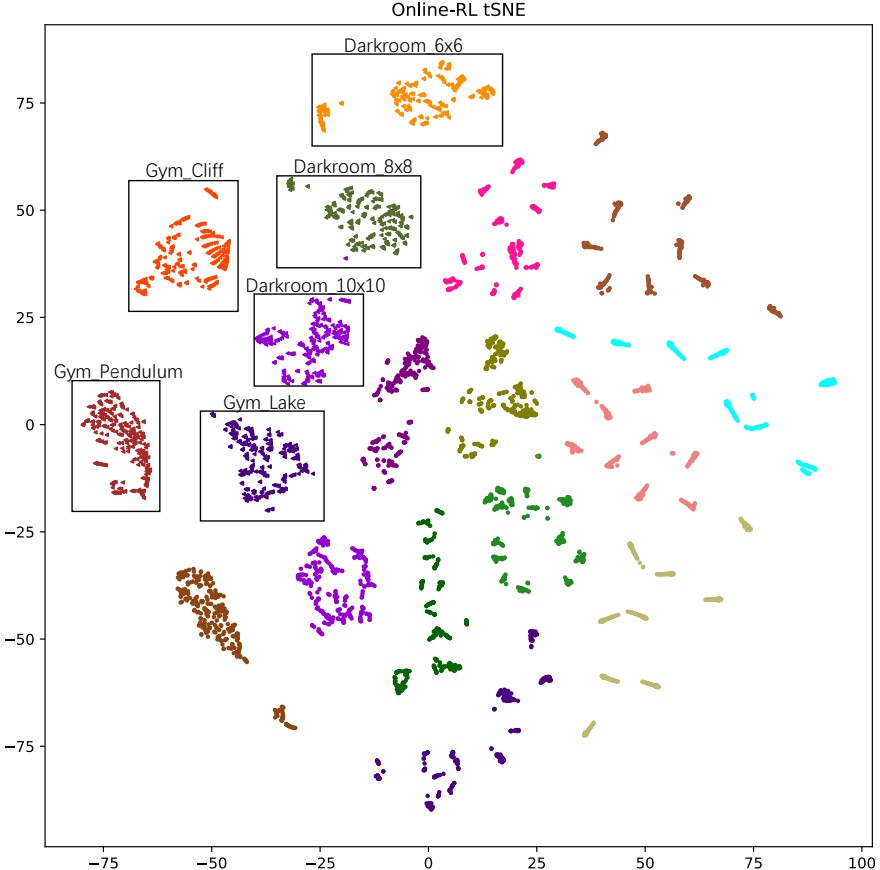

Figure 18: t-SNE visualization of the trajectory of the memory states ($\phi_t$) of OmniRL in online-RL evaluation with variant environments. The unboxed points correspond to $\mathcal{T}(16, 5)$. Trajectories originating from the same environment are represented in the same color.

Table 8: Comparison of the performance (success rate) of LLM and other methods discussed in the paper within the *FrozenLake* environment.

| METHODS | NON-SLIPPERY | SLIPPERY |
|---|---|---|
| RANDOM | 1.6% | 1.3% |
| LLM W/ STATE ONLY | < 2% | < 2% |
| LLM W/ GLOBAL MAP | 100% | 5.6% |
| LLM W/ GLOBAL MAP & HINT | | 17% |
| OMNIRL W/ STATE ONLY | 100% | 60% |
| ORACLE | 100% | 75% |

## D.5 Comparison with Pre-trained LLMs

We also investigate whether a well-pretrained LLM can naturally solve the decision-making tasks investigated in this paper. To circumvent the lack of common sense in AnyMDP tasks, we primarily conducted tests in the *FrozenLake* task with *DeepSeek-R1*[78] in two modes:

1. Similar to the evaluation of standard ICRL, we do not provide the agent with the map. Instead, we report only the state ID and reward of the agent. The initial prompts used to initiate the evaluation are shown in Figure 19.

You are playing the Frozen Lake game. The environment is a 4x4 grid where you need to maximize the success rate by reaching the goal (+1) without falling into holes (-1). You can move in four directions: left, down, right, and up (represented as 0, 1, 2, 3 respectively). You will receive the current state and need to provide the optimal action based on your learning. When asked for the optimal action, your response must be an integer ranging from 0 to 3, and no other context is permitted. There are two kind of request types:

1.integer: the integer is the current state, and you need to provide the optimal action.

2.list: The list contains one or more tuples, where each tuple contains the last state, action taken, reward received, and next state. To save time, you don't need to respond when receiving a list. You will play the game multiple times. A game ends when the reward is -1 or 1, try to get a higher success rate.

Note: I am asking you to play this game, not to find a coding solution or method.

You will be provided with a conversation history. The latest prompt is the current state, and others are the list of sequential environment feedback history in tuple type. Each tuple contains four values, the first one is state, the second one is action, the third one is reward and the fourth one is next state.

Your response must be an integer from 0 to 3 during the entire chat.

If you find the last state is equal to the next state, your policy in the last state can't be this action.

If you find the reward in the tuple is -1, your policy in the last state can't be this action.

You need to get to the goal as soon as possible.

Figure 19: Prompts for LLM to initialize the Lake4 × 4 (Slippery) task without a global map

There is a game with the following basic description and rules:

Frozen Lake involves crossing a frozen lake from the start to the goal without falling into any holes by walking over the frozen lake. The player may not always move in the intended direction due to the slippery nature of the frozen lake.

The game starts with the player at location [0,0] of the frozen lake grid world, with the goal located at the far extent of the world, for example, [3,3] for the 4x4 environment.

Holes in the ice are distributed in set locations when using a pre-determined map or in random locations when a random map is generated.

The player makes moves until they reach the goal or fall into a hole.

The lake is slippery, so the player may move perpendicular to the intended direction sometimes. If the intended direction is to the left, the actual move may be to the left, up, or down, with the corresponding probability distribution: P(move left) = 1/3, P(move up) = 1/3, P(move down) = 1/3. If the intended direction is to the right, the actual move may be to the right, up, or down, with the corresponding probability distribution: P(move right) = 1/3, P(move up) = 1/3, P(move down) = 1/3. If the intended direction is up, the actual move may be up, left, or right, with the corresponding probability distribution: P(move up) = 1/3, P(move left) = 1/3, P(move right) = 1/3. If the intended direction is down, the actual move may be down, left, or right, with the corresponding probability distribution: P(move down) = 1/3, P(move left) = 1/3, P(move right) = 1/3. You are given a 4x4 map where:

S represents the start.

F represents the frozen surface that can be walked on.

H represents a hole; falling into it will return the player to the start.

G represents the goal.

The map is as follows:

The first row from left to right is "SFFF".

The second row from left to right is "FHFH".

The third row from left to right is "FFFH".

The fourth row from left to right is "HFFG".

Please determine the optimal policy that maximizes the success rate of safely reaching the goal from the start. The optimal policy is the intended direction at each map location, where actions 0, 1, 2, and 3 represent moving left, down, right, and up, respectively.

Note: You are not required to write code to solve this problem; instead, directly provide the optimal policy.

Figure 20: Prompts for LLM to solve Lake4 × 4 (Slippery) with global map

In this particular environment, you should focus more on strategies to avoid encountering pitfalls rather than concentrating on reaching the goal. A reevaluation of the decision-making process in this context may yield more effective outcomes.

Figure 21: Additional guide (hint) for LLM to solve Lake$4 \times 4$ (Slippery) with global map

The optimal policy for each state (row, column) is as follows, where each cell lists the intended direction (action) that maximizes the probability of reaching the goal:
- **(0,0)**: Down (1) - **(0,1)**: Down (1) - **(0,2)**: Down (1) - **(0,3)**: Up (3) - **(1,0)**: Down (1) - **(1,2)**: Down (1) - **(2,0)**: Right (2) - **(2,1)**: Down (1) - **(2,2)**: Down (1) - **(3,1)**: Right (2) - **(3,2)**: Right (2)
**Explanation**: - **Top row (0,0 to 0,3)**: Prioritize moving down to avoid the risky rightward path toward holes. - **Middle rows (1,0 and 1,2)**: Move down to progress toward safer lower rows. - **Lower rows (2,0 to 3,2)**: Navigate rightward toward the goal, avoiding holes. For example, (3,1) and (3,2) move right to reach the goal directly. - **(0,3)**: Move up (intended) to bias movement leftward (safer) and avoid falling into the hole at (1,3).

Figure 22: Responses from DeepSeek-R1 for Lake$4 \times 4$ (Slippery) with global map

The optimal policy, revised to prioritize avoiding holes, is as follows:
- **(0,0)**: Down (1) *Avoids (0,1)'s risks by favoring transitions to (1,0) or (0,1) without direct hole exposure.*
- **(0,1)**: Right (2) *Moves toward (0,2) or (1,1) (hole). While risky, all actions here have 1/3 hole risk; right progresses toward safer (0,2).*
- **(0,2)**: Right (2) *Directs to (0,3) instead of down to (1,2), which is high-risk.*
- **(0,3)**: Up (3) *Biases movement left (to (0,2)) and avoids the hole at (1,3).*
- **(1,0)**: Left (0) *Avoids 1/3 chance of moving right into (1,1) (hole). Transitions to (0,0) or (2,0) safely.*
- **(1,2)**: Right (2) *Despite 1/3 hole risk, it allows progress to (2,2) or (0,2). Left (action 0) has equal risk but less progress.*
- **(2,0)**: Up (3) *Avoids 1/3 chance of moving down into (3,0) (hole). Routes to (1,0) or (2,1).*
- **(2,1)**: Down (1) *Safely moves to (3,1) (frozen), progressing toward the goal.*
- **(2,2)**: Down (1) *Reaches (3,2), a safe step toward the goal.*
- **(3,1)**: Right (2) *Directly progresses to (3,2), then (3,3) (goal).*
- **(3,2)**: Right (2) *Final step to the goal (3,3).*
### Key Adjustments from Original Policy: 1. **(1,0)**: Changed to **left** (0) to avoid 1/3 chance of falling into (1,1). 2. **(1,2)**: Changed to **right** (2) instead of down, reducing hole risk from 2/3 to 1/3. 3. **(2,0)**: Changed to **up** (3) to avoid 1/3 chance of entering (3,0). 4. **(0,1)**: **Right** (2) minimizes repeated exposure to (1,1) compared to down.
### Path Safety Focus: - Avoids actions with >1/3 hole probability (e.g., down from (1,2)). - Prioritizes paths with lower immediate risk, even if longer (e.g., routing through (2,0) → (2,1) → (3,1) → goal).

Figure 23: Responses from DeepSeek-R1 for Lake$4 \times 4$ (Slippery) with global map and hint

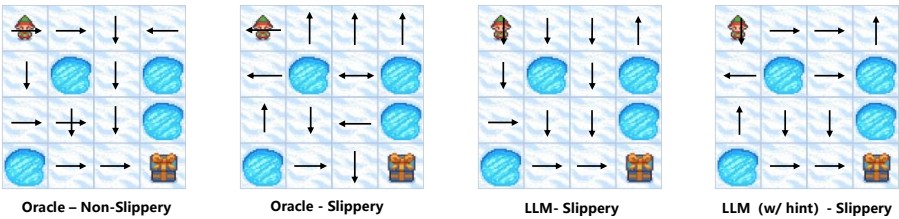

Figure 24: Comparison of the solutions of different methods in the *FrozenLake* environment. In the LLM w/ hint condition, we provide additional guidance to the agent, instructing it to prioritize avoiding holes over reaching the goal.

2. We initially provide the global map to the $DeepSeek - R1$ and then commence the interaction. In this mode, the LLM can leverage the global map to make decisions. The prompts are shown in Figure 20 and Figure 21

As shown in Table 8 and Figure 24 (results executed by following the responses of DeepSeek-R1 (version 2025/03) in Figure 22 and Figure 23), LLM agents are only able to solve the *FrozenLake (non-slippery)* environment when provided with a global map. Without access to a global map, we conducted extensive interactions between LLM agents and the environment, running up to 500 episodes (100, 000 steps). Despite these efforts, the agents failed to solve even the non-slippery variant of the task, achieving scores that were comparable to those of a random policy.

Even with the aid of a global map, the performance of LLM agents on the *FrozenLake (non-slippery)* environment remains notably poor. To improve their performance, we introduced additional hints suggesting that a better solution should prioritize avoiding holes over reaching the goal. However, this intervention only marginally improved the agents' performance, raising it from 5.6% to 17%. This level of performance is still significantly lower than that of the Oracle and OmniRL agents.

Notably, increasing the length of the chain of thought can potentially enhance performance when a global map is available, but it has minimal impact on performance when only the current state is considered. The former scenario emphasizes System 2 decision-making, which is characterized by rule-based and analytical thinking. In contrast, the latter scenario highlights the in-context adaptation of System 1 decision-making, which relies on continual external feedback and represents rapid, intuitive decision-making [79, 80]. We argue that future research should place greater emphasis on the latter approach.

