# OpenReview forum: "Towards Large-Scale In-Context Reinforcement Learning by Meta-Training in Randomized Worlds"
_NeurIPS.cc/2025/Conference — NeurIPS 2025 poster_

### Official Review · Reviewer_Dwj8 · 2025-06-27

**Clarity:** 3
**Significance:** 3
**Originality:** 3
**Rating:** 5
**Confidence:** 4

**Summary:**

In this paper the authors introduce AnyMDP, a novel generation framework that is able to create a wide range of tabular Markov Decision Processes (with distinct state-action spaces, transition functions and reward functions) to support research in in-context reinforcement learning (ICRL). Furthermore, the authors propose a training methodology for meta-learners that employs AnyMDP. The resulting model, "OmniRL" is evaluated in unseen tasks, against online RL approaches (Tabular Q-Learning with UCB action selection, PPO) trained on those tasks. The authors highlight how OmniRL performs on par with these approaches while requiring significantly less interactions/data. Subsequently, the authors highlight how OmniRL outperforms other in-context learning methods. Subsequently, the authors evaluate the role of the number of tasks, training iterations and context lenght in the performance of their method.

**Questions:**

1) Can the authors elaborate on what algorithms were used to generate the behavior policies (line 181)? For example, what model-based algorithm was employed? It would benefit the paper if the authors could describe how they were trained in Appendix. Also, are the Q-learning and model-based behavior policies always optimal, or they have different degrees of optimality?
2) In the sampling process of MDPs how do you make sure that the MDPs are ergodic?
3) Have the authors attempted to evaluate their approach in out-of-distribution tasks in regards to number of states and number of actions? Especially in tasks with more states and actions.
4) I found the results with the multi-agent task quite interesting but the discussion in the text was quite unclear. Can the authors elaborate on what does "configuring observation spaces" mean (line 236)? Are the policies of the other agents fixed, or learned as well? If fixed, then the task essentially reduces to a single agent task, with the actions of the other agents part of the "environment" transition function. A short description of these tasks in Appendix would also clarify other future readers of the paper.
5) In Section 4.2. the authors evaluate their approach without a priori knowledge, as an ablated version of their method. What about the performance of OmniRL without SS, using for example AD, AD$^\epsilon$ and DPT, but with priori information?

**Ethical Concerns:**

["NO or VERY MINOR ethics concerns only"]

**Final Justification:**

The authors introduce a novel generation framework that is able to create a wide range of tabular Markov Decision Processes (with discrete state/action spaces) and show that it enables in-context learning of unseen MDPs. I believe this contribution is both novel and relevant for the RL community. The authors also addressed positively my questions and concerns. As such I maintain my positive score.

**Limitations:**

yes

**Paper Formatting Concerns:**

None.

**Quality:**

4

**Strengths And Weaknesses:**

**Strengths**:
- This paper presents a strong contribution for in-context reinforcement learning: a novel generation framework for tabular MDPs and a novel training methodology for ICRL.
- Extensive experimental results, showing strong performance and novel insights into the performance of ICRL.
- Well written and clear paper.

**Weaknesses**:
- Some missing details and minor typos (See "Comments" below)


**Comments**:
- Overall, I thoroughly enjoyed this work. The authors did a good job motivating the need for this benchmark, and how it differs from previous approaches. The writing is clear and the paper is well structured.
- Some variables are not defined in the text: $\epsilon_P, \epsilon_R$ (line 132), $w_t$ (line 4).
- The captions of the figures could be improved and expanded, in particular Figure 1 and Figure 2. I understand that the authors were constrained by the space available, but I highly incentivize the authors to expand the captions in a revised version. Furthermore, Figure 2 is of low quality, contrasting with the rest of the paper: I would incentivize the authors to make a more complete figure, even if in Appendix, that explains more in detail the training scheme. The legend of Figure 5 is also too small and unable to be read in the printed version).
- The banded transition kernel and composite reward functions introduced in Section 3.1 need to be discussed (at least in part) in the main paper. I understand the constrains with available space in the paper, but these are fundamental components to understand the sampling process of tasks in the AnyMDP and, as such, should not be fully relegated to Appendix.
- The authors also use the concept of "burstiness" throughout the paper (line 76, line 289, line 290, line 316) yet do not define it. For clarity, it would be better to define it in the text, instead of relegating the definition to a reference.
- The acronym ICRL is defined in two different ways: "in-context reinforcement learning" and "inverse contextual reinforcement learning" (line 309). It is better to be consistent.
- There is an extra " in line 256 that should be removed.

---

> ### Author Rebuttal · Authors · 2025-07-29
>
> First and foremost, we greatly appreciate the reviewer for pointing out the typos in the comments, as well as the constructive suggestions regarding paper writing. We will revise accordingly to further improve the paper’s readability and quality. In the following, we will elaborate in detail on the questions raised by the reviewer.
>
> # Questions on AnyMDP
> ## Behavior policies
> (To answer the reviewer's Question 1.)
> ### Definitions for each policy
> On page 20, we present Table 5 about "Correspondence of prompt IDs and the policies it represents". We thank the reviewer for pointing out the issue of lacking exact definitions for each policy, and will subsequently include them in the paper. For IDs 0 to 3, the policies are value iteration policies with different gamma factors, corresponding to short-term to long-term oracles. For the model-based reinforcement learner, we implement the RL solver algorithm based on Optimistic Thompson Sampling (OTS) for episodic reinforcement learning problems in unknown environments. It balances exploration and exploitation by dynamically adjusting exploration noise (incorporating UCB heuristics) to learn the optimal policy. The Tabular Q Learner is a model-free reinforcement learning algorithm that maintains a Q-table to store Q-values for each state-action pair, updates these values via temporal-difference learning using rewards and next-state information from environmental interactions.
>
> ### Purpose of using different learning methods
> While the Q-learning and model-based behavior policies are always optimal at the end of training, their learning methods and paths during training differ. In our data generation process, these two policies learn synchronously along the trajectory of the data, rather than using pre-trained models to generate data. Because what we aim for our OmniRL model to learn is how to converge online to a good policy in a new task through interaction with the environment. Thus, the learning processes of these two behavior policies are actually more important than the final optimal policy—after all, the oracle policy can be directly obtained via value iteration.
>
> ## Random sampling with condition ensures both diversity and quality
> (To answer the reviewer's Question 2.)
>
> The sampling of transition matrix and reward matrix in AnyMDP is not uniformly random, to avoid generating trivial MDP problems (e.g., MDPs with optimal episode length=1). Uniform randomization would result in generating a majority of trivial Markov Decision Processes (MDPs), such as those with optimal episode lengths of 1 or MDPs where most states are irrelevant to the optimal solution. The policy only needs to remember which short path has the best reward and will not learn the trade-off of risk assessment. Eventually, the environment will be more inclined towards the bandit problem. In most complex RL problems, delayed reward is quite common. Usually, a large positive reward is only obtained when the goal is achieved. Sometimes, there may even be a stage of deduction before getting the positive reward.
>
> Ergodicity is ensured by Theorem 1 in the appendix, which guarantees that a uniformly random policy induces a strictly positive probability of reaching every state from the initial state. At the same time, AnyMDP does not guarantee that the optimal policy is strictly ergodic—a trait absent in most existing benchmarks. However, the banded transition kernel and reward structure ensure that the optimal policy requires transitions across a substantial number of states.
>
> Thus, via the "Banded transition kernel" condition specified in Line 525, we ensure the path to reaching high-value targets is longer. Following the procedure outlined in Algorithm 1 (page 15), we: randomly generate states with varying values (Line 2 of Algorithm 1); randomly select states that trigger reset or termination (Lines 3–4); randomly sample transition matrices under the "Banded transition kernel" condition (Lines 5–12); and randomly sample composite reward functions that satisfy the value condition (Lines 13–19). This approach ensures both task diversity and quality simultaneously.
>
> ## Priori knowledge
> (To answer the reviewer's Question 5.)
>
> The reviewers raised questions regarding the inclusion of prior information in AD, AD𝜖, and DPT. To clarify, prior information introduces additional details about different behavioral policies—specifically, policy tags. Since the AD policy remains unchanged during generation, adding prior information has no theoretical effect. For AD𝜖, policy tags can be incorporated based on the noise level. Additionally, as DPT involves multiple policies, the inclusion of policy tags is feasible. We believe that adding policy tags is also highly likely to enhance performance for AD𝜖 and DPT.
>
> Indeed, compared with DPT, our primary contribution lies in proposing the Step-wise Supervision method, which enhances the strength of supervisory signals and improves training efficiency. As for policy tags, they serve as an additional benefit—analogous to high-quality prompts in LLMs—by informing the model of the actual level/type of the current action, thereby facilitating strategy distillation during training.
>
> # Question on experiment settings
> ## Testing task
> (To answer the reviewer's Question 3.)
>
> We chose the RL benchmark Gym for our out-of-distribution tasks. In most suitable games within it, state and action spaces are fixed, making it infeasible to conduct a comprehensive experiment on the number of states and actions. However, in Figure 3 on page 6, we compare performance across AnyMDP tasks with different numbers of states, where results show that difficulty increases as the state space grows.
>
> ## Multi-agent test
> (To answer the reviewer's Question 4.)
>
> Both agents are independent OmniRL agents. They perform ICL in the same environment with distinct observations. Observations are partial: one agent can only observe the column number of the other agent. Thus, both agents operate dynamically. We were also pleasantly surprised that the model—trained on single-agent datasets with a single-agent architecture—could solve multi-agent problems. This also inspires us to explore the application of ICL in multi-agent scenarios.

---

> > ### Comment · Reviewer_Dwj8 · 2025-08-01
> > **Acknowledgment of Author's rebuttal**
> >
> > Dear authors,
> >
> > Thank you for the rebuttal and for clarifying my questions! For now, I maintain my positive score.
> >
> > I would also incentivize the authors to (in the future) run some experiments (even if simple ones) on 2D maze-like environments of different size to test the extrapolation of the method to scenarios of increased number of states. For increased number of actions, the authors can also explore higher-dimensional mazes.

---

> > > ### Author Response · Authors · 2025-08-04
> > > **Follow-up**
> > >
> > > We sincerely appreciate the reviewer’s thoughtful feedback and constructive suggestions. We are particularly encouraged by their recognition of our work’s potential and are already advancing the application of ICRL to complex domains such as 3D navigation and locomotion in separate ongoing studies. Notably, several methodologies presented in this paper have undergone preliminary cross-validation through these parallel efforts, reinforcing their robustness and adaptability. We look forward to incorporating the reviewer’s insights to further refine these extensions in future work.

---

### Official Review · Reviewer_zr1Z · 2025-07-02

**Clarity:** 3
**Significance:** 3
**Originality:** 3
**Rating:** 3
**Confidence:** 5

**Summary:**

The paper introduces the AnyMDP framework, which utilizes randomized Markov Decision Processes (MDPs) to generate diverse tasks for meta-training in In-Context Reinforcement Learning (ICRL) at a large scale. By incorporating step-wise supervision and prior knowledge induction, the framework improves efficiency and generalization, demonstrating that OmniRL, trained on these randomized tasks, performs well across unseen environments. The work highlights the importance of task diversity and world randomization in scaling ICRL and achieving robust performance across different tasks. The experiments show that larger task sets lead to better generalization but at the cost of increased adaptation time.

**Questions:**

Q1: The paper predominantly focuses on discrete tasks; however, many real-world environments are continuous or partially observable. How do the authors intend to extend the proposed framework to handle such complex, continuous, or partially observable environments?

Q2:  The paper proposes the AnyMDP framework, but its effectiveness has yet to be comprehensively evaluated across a wider range of in-context reinforcement learning methods. It would be valuable for the authors to assess and demonstrate its applicability in this broader context to strengthen the validity of their claims.

Q3: The proposed framework has not been tested across a broad range of in-context reinforcement learning tasks. How do the authors plan to evaluate the generalizability of the framework across diverse in-context RL environments, particularly those with varying complexity and different types of disturbances? Would additional benchmarks or real-world applications be considered to assess its robustness in such settings?

Q4: The paper mentions that perturbations can lead to a decline in model performance, but it does not delve into the specific impacts of different types of perturbations (such as observation noise, action perturbations, and reward errors) on the model. How are these diverse perturbations simulated in AnyMDP, and how is the model's robustness ensured across various environmental changes?

Q5: The computational costs of generating large-scale tasks and training the model are not fully discussed. Can you elaborate on the computational resources required for training on a large number of tasks, especially when task complexity increases?

**Ethical Concerns:**

["NO or VERY MINOR ethics concerns only"]

**Final Justification:**

In summary, I appreciate the paper’s scope of ICRL scalability and its task distribution construction with structured diversity. My main concern is that the empirical evidence might be insufficient to support the key claim on large-scale ICRL, which was not well addressed by the authors’ response. I tend to maintain my current score as the reasons to reject outweigh the reasons to accept. If with sufficient empirical evidence on large-scale problems, I believe this paper is well qualified for the next venue.

**Limitations:**

Yes

**Quality:**

2

**Strengths And Weaknesses:**

### Strengths

The paper's major strengths include its scalability, with AnyMDP enabling large-scale task generation, and the world randomization approach that reduces structural bias in task sets, leading to strong generalization across new environments. The use of step-wise supervision and prior knowledge induction also enhances the efficiency of meta-training.

### Weaknesses

The paper’s main weaknesses include its limited applicability to real-world continuous or partially observable environments, as it focuses on discrete tasks. The computational costs of generating and training on large-scale tasks are not fully discussed. While task diversity is emphasized, the performance trade-offs with smaller task sets are not explored, and the longer adaptation times with larger task sets may limit real-time applications. Additionally, the paper doesn't sufficiently address the challenges of long-term generalization and knowledge accumulation in dynamic environments.

---

> ### Author Rebuttal · Authors · 2025-07-30
>
> # On the Value of Discrete Space MDPs
> (To answer the reviewer's Question 1.)
>
> We appreciate suggestions to extend our framework to continuous MDPs, but we feel that the value of discrete MDPs is underestimated. At this stage, partially pursuing the task complexity risks undermining the core advantages of procedurally generated discrete MDPs. Below, we explain the challenges of preserving key properties in continuous domains and contrast our approach with contemporary benchmarks.
>
> ## Challenges of Continuous MDPs in Preserving Structural Integrity
> Extending procedurally generated MDPs to continuous spaces while maintaining low structural bias and sufficient diversity poses significant technical hurdles. At the current stage, efforts moving to continuous MDPs risk typically **prioritizing "absolute task complexity" over qualitative diversity**, introducing biases or homogenizing task structures—contrary to our framework's goals.
>
>
> ## Inherent Long-Term Dependency and Qualitative Diversity
> (To answer the reviewer's Question 3.)
>
> The reviewer suggests conducting tests on tasks with varying complexity and different types of disturbances. In reality, existing continuous MDP benchmarks rely on continuously randomized hyperparameters to simulate "infinite tasks," prioritizing few-shot learning over long-horizon in-context learning (ICL), such as environments like maze navigation and open-world games. Our analysis reveals two insights:
> - Task Diversity ≠ Numerical Scale: True diversity depends on qualitative variation in task structures, not just quantity.
> - ICL efficacy depends on disparity between tasks in the training set, not individual task complexity. Existing continuous benchmarks tend to produce homogeneous task structures, limiting models to superficial pattern recognition via in-weight learning (IWL) rather than sustained reasoning. In contrast, discrete MDPs in AnyMDP are designed with procedural complexity and temporal coherence, enabling rigorous evaluation of a model’s ability to generalize across structurally distinct tasks and sustain reasoning over extended sequences.
>
> ## Compatibility with Existing LLMs
> Beyond being a novel benchmark, AnyMDP bridges a critical gap in assessing experience-driven, holistic reasoning capabilities of pre-trained LLMs. Existing benchmarks predominantly measure declarative memory and one-shot circuits, while AnyMDP emphasizes procedural memory and experience-driven learning.
>
> ## Low-cost Access to Oracle Policy
> This property significantly enhances the efficiency and scalability of ICRL training, making AnyMDP a scalable benchmark.
>
> ## Empirical analysis across state-of-the-art architectures
> We augment the empirical analysis with the training curves and ICL comparisons across state-of-the-art architectures, including Mamba2, RWKV-7, Gate Delta Net, and other competitive models. This will demonstrate AnyMDP’s utility as a rigorous benchmark for advancing long-context modeling capabilities.
>
> - Table 1. Validation Loss of Different Models on AnyMDP Datasets
> | CL 	  | 1  	   | 100	    | 1000    | 10000   | 30000 |
> |-------------|------------|------------|------------|------------|------------|
> |  Mamba2   | 1.417 ± 0.035 | 1.272 ± 0.051 | 0.822 ± 0.064 | 0.436 ± 0.053 | 0.294 ± 0.042 |
> |  GSA   | 1.406 ± 0.034 | 1.235 ± 0.053 | 0.765 ± 0.066 | 0.342 ± 0.050 | 0.235 ± 0.042 |
> |  Gate DeltaNet   | 1.356 ± 0.035 | 1.130 ± 0.051 | 0.599 ± 0.055 | 0.267 ± 0.049 | 0.173 ± 0.042 |
> |  RWKV-7   | 1.371 ± 0.034 | 1.151 ± 0.052 | 0.627 ± 0.055 | 0.269 ± 0.042 | 0.175 ± 0.029 |
>
>
> Overall, in our research on ICL, the benefits brought by the discrete MDPs far outweigh the drawbacks of its limited problem space. We have also started to explore ICL in the continuous space.
>
> For future work in continuous settings, there are two directions. First, we will focus on specific continuous problems and prioritize in-domain adaptation. For instance, in frame prediction tasks, generalization manifests as adaptation to environments with distinct features and spatial structures. Similarly, in robot whole-body locomotion control, it manifests in adapting to different robot configurations (size, shape, weight). Second, we will continue exploring how to generate tasks with high diversity and low structural bias in continuous spaces. Undoubtedly, the first direction holds more practical value, while the second poses greater theoretical challenges.
>
> # The "Perturbation" in Question 4
> The reviewer mentions "perturbations can lead to a decline in model performance." We believe there are no such claims in our paper. Two candidates appear in our manuscript, neither fitting this question:
> - Domain/World randomization (Line 132): This creates task diversity; AnyMDP already exposes the learner to maximal variations, so no need to delve into specific impacts of perturbations.
> - Stochasticity in the behavior policy (Line 620 & Table 5): This is necessary for data synthesis and seems irrelevant to the mentioned perturbation. Moreover, incorporating Domain/World randomization and stochasticity in the behavior policy has enhanced model performance, contradicting the reviewer's viewpoint.
> # About Long-Term Adaptation in Weaknesses
> The reviewer views long-term adaptation as a disadvantage of our methods, but we believe there are misunderstandings. Our proposed methods, step-wise supervision (SS) and chunk-wise learning, aim to improve in-context sample efficiency, not intentionally extend context length. Our experiments reveal that **"Long-Term Adaptation is a tax we must pay toward increasing the generalization scope for ICRL."** Characterizing increased context length as an inherent limitation seems unjustified, even from the standpoint of whether trading generalization for long-context adaptation is worthwhile.
>
> # About Computational Costs in Question 5
> In Figure 7 on page 17, we show the time consumption of the AnyMDP task generation on an Intel(R) Xeon(R) Platinum 8374C
> CPU. After generating the task files, the time consumption of generating the sequences accordingly is within hours, which can be ignored compared to the training cost. To train an OmniRL model with [16,128] state space and action space of 5, we generate 512K sequences with length 12K. The total training iterations are 0.2M, 8 seconds per iteration on 8 GPUs. We have also considered conducting the training across task complexity to assess the exact consumption, but the computing resources are tight. In general, training a model with a larger number of states or actions will bring greater resource consumption. After the problem becomes more complex, more sequences are needed to learn generalization, and each sequence has to be longer to obtain sufficient information for a single task.
>
> # Evaluated More ICRL Methods in Question 2
> We have conducted experiments on well-known ICRL frameworks, including AD, AD𝜖, and DPT. Another technical direction is RL². However, with existing large-scale task sets, model scales, and context lengths, the computational cost of running RL is substantial. Using fewer tasks, smaller models, or shorter context lengths would render the conclusions lacking in comparative significance. Furthermore, methods like DPT have already been compared with RL²; thus, we believe the results from comparisons with DPT are sufficiently compelling.

---

> > ### Comment · Reviewer_zr1Z · 2025-08-06
> >
> > Thank the authors for their detailed response. I am on the edge now.
> > - On the positive side, the scope of the paper is very broad, addressing the scalability of ICRL that is the future trend of RL foundation model development. Also, the way of constructing a large distribution of training tasks with structured diversity is on the right direction to train large-scale models towards high generalization capacities.
> > - On the negative side, the proposed scheme can only work for discrete state-action space, which is easily limited by the problem complexity. This is on the contrary to the development of RL scalability. It remains unclear whether the principles of constructing the training set and meta-training can be extended to general RL settings with continuous state-action spaces. Also, the commonly adopted ICRL benchmarks (e.g., MuJoCo and Meta-World) are not investigated in the paper, posing a barrier to assesssing its applicability and advantages in ICRL domains.
> >
> > I am willing to hear more justifications from the authors and discussions from other reviewers.

---

> > > ### Author Response · Authors · 2025-08-07
> > >
> > > We sincerely thank the reviewers for their constructive feedback and insightful suggestions. We acknowledge the observation that discrete state-action spaces may initially seem limited in scope, and we are excited to share that we are actively submitting separate manuscripts on ICRL for tasks including navigation and locomotion. To further clarify the core contribution of this work, it mainly addresses the scalability issue of ICRL, and demonstrates the potential of multi-paradigm ICL:
> > >
> > > - Scalable Task Set:
> > > AnyMDP addresses a critical gap by enabling scalable, low-bias benchmarking for ICRL. By imposing only a minimal structural assumption—discreteness—it ensures flexibility while avoiding domain-specific inductive biases.
> > >
> > > - Scalable Training:
> > > Similar to Chain-of-Thought (CoT) Reasoning, ICRL requires sophisticated training that depends on Reinforcement Learning (e.g., RL^2), which is challenging to scale. Existing distillation methods (e.g., AD, ND) are inefficient according to our validation in AnyMDP, which is far more diverse and on a much larger scale than previous experiments. While the proposed Stepwise Supervision (SS) method is inspired by DPT and DAgger, DPT’s (trajectory, label) data format is far less efficient than SS. We position SS as a promising direction for resolving scalability challenges, which makes the training of ICRL available to the scale of pretraining.
> > >
> > > - Multi-Paradigm ICL:
> > > We demonstrate that integrating prior information into ICRL enables multi-paradigm ICL across RL, Offline-RL, and Imitation Learning. This expands the scope of meta-RL and ICRL to align with broader general-purpose ICL paradigms, suggesting new avenues for unified Multi-Paradigm ICL.
> > > We will further include some of those insights into the revision.
> > >
> > > Additional clarifications on the task set:
> > > - On Continuous Environments: Current continuous benchmarks are typically confined to single-domain tasks or multi-domain setups comprising fragmented, isolated fields. Constructing general-purpose, structurally unbiased continuous environments remains highly challenging and computationally intensive. Discrete frameworks offer a pragmatic alternative that balances tractability with expressiveness.
> > > - Applicability Beyond Discreteness:
> > >     - Discrete MDPs can effectively approximate certain continuous problems (e.g., our experiments with discretized pendulum and mountain car tasks).
> > >     - A promising extension involves multi-discrete-token state/action spaces, which align with modern trends in AI (e.g., VLM, VLA). This could unlock broad applicability across diverse domains.

---

> > > > ### Comment · Reviewer_zr1Z · 2025-08-09
> > > >
> > > > Thank the authors for the new response. I am more interested in the authors’ work on ICRL for continuous tasks including navigation and locomotion. Maybe the key concern I am hesitating is not about discrete to continuous state-action spaces, but the required problem complexity to support the paper’s claim on “large-scale ICRL”. As we know, the language problem also involves a discrete space, but the space is extremely large due to a large vocabulary (e.g., 150k) and a long-horizon sequence (e.g., 8k). In contrast, the problems involved in the paper are with small state-action spaces, less than 128 states and 5 actions. As I appreciate the scope of this paper being the scalability of ICRL that is the future trend of RL foundation model development, it may be necessary to validate the claim on bigger problems, demonstrating the scalability w.r.t. bigger model structure and higher problem complexity. I suggest the authors to conduct empirical evaluate on more complex discrete problems, e.g., discretizing a complex continuous domain with higher precision.
> > > >
> > > > In summary, I appreciate the paper’s scope of ICRL scalability and its task distribution construction with structured diversity. My most concern is that the empirical evidence might be insufficient to support the key claim on large-scale ICRL, which was not well addressed by the authors’ response. I tend to maintain my current score as the reasons to reject outweigh the reasons to accept. If with sufficient empirical evidence on large-scale problems, I believe this paper is well qualified for the next venue.

---

### Official Review · Reviewer_WkeX · 2025-07-02

**Clarity:** 2
**Significance:** 3
**Originality:** 3
**Rating:** 4
**Confidence:** 3

**Summary:**

The paper studies in-context learning for reinforcement learning, contributing a task generation framework called AnyMDP that generates tabular MDPs with randomized transition and reward functions. It proposes several techniques to help with meta-training in such randomized worlds.

**Questions:**

1. Can the authors shed some more light on why their sampling procedure (as detailed in Appendix B.1) results in high-quality MDPs by providing more intuition on why these are considered to be of high quality?

2. Did the authors consider using the Deep Sea environment, which is a well-known sparse reward benchmark? More generally, how were the presented Gymnasium environments selected?

3. What was the impact of the reward shaping? Specifically, what is the performance of the model is if is not aided by reward shaping?

**Ethical Concerns:**

["NO or VERY MINOR ethics concerns only"]

**Final Justification:**

The authors have satisfactorily answered my questions. I’m confident the responses can be easily incorporated in a next version of the paper, leading me to raise my score.

**Limitations:**

An obvious limitation that the authors readily acknowledge in Section 5 is the fact that the framework is limited to discrete action and state spaces. Some more discussion about how the work could be extended to continuous domains would be appreciated.

**Paper Formatting Concerns:**

None.

**Quality:**

2

**Strengths And Weaknesses:**

Strengths
1. Highly relevant and timely topic
2. Good positioning in related work
3. Promising experimental results but with some caveats

Weaknesses
1. Some discussion on the procedural generation algorithm would be valuable, focusing on why the objective as presented on lines 143/144 was selected (for instance by detailing some of the analysis referred to on line 140).
2. Reward shaping was applied in standard benchmark domains (see Questions). The selection of domains is unclear.
3. The exact definition of the 7 types of behavior policies should be clarified, as well as the exact impact of diversity.
4. On line 132, the two epsilon terms appear to be undefined.

---

> ### Author Rebuttal · Authors · 2025-07-29
>
> First and foremost, we sincerely thank the reviewer for identifying the typos. We will revise accordingly to further improve the paper’s readability. In the following, we will elaborate in detail on the weaknesses and questions raised by the reviewer.
>
> # On the Value of Discrete Space MDPs
> We appreciate the reviewer's interest in extending the task to continuous domains. Before discussing how to extend to continuous spaces, we would first like to engage in a deeper discussion with the reviewers about why we chose discrete MDPs at this stage, as well as the benefits this choice brings.
>
> At this stage, partially pursuing the task complexity risks undermining the core advantages of procedurally generated discrete MDPs. Below, we explain the challenges of preserving key properties in continuous domains and contrast our approach with contemporary benchmarks.
>
> ## Challenges of Continuous MDPs in Preserving Structural Integrity
> Extending procedurally generated MDPs to continuous spaces while maintaining low structural bias and sufficient diversity poses significant technical hurdles. At the current stage, efforts moving to continuous MDPs risk typically **prioritizing "absolute task complexity" over qualitative diversity**, introducing biases or homogenizing task structures—contrary to our framework's goals.
>
>
> ## Inherent Long-Term Dependency and Qualitative Diversity
> Existing continuous MDP benchmarks rely on continuously randomized hyperparameters to simulate "infinite tasks," prioritizing few-shot learning over long-horizon in-context learning (ICL). Our analysis reveals two insights:
> - Task Diversity ≠ Numerical Scale: True diversity depends on qualitative variation in task structures, not just quantity.
> - ICL efficacy depends on disparity between tasks in the training set, not individual task complexity. Existing continuous benchmarks tend to produce homogeneous task structures, limiting models to superficial pattern recognition via in-weight learning (IWL) rather than sustained reasoning. In contrast, discrete MDPs in AnyMDP are designed with procedural complexity and temporal coherence, enabling rigorous evaluation of a model’s ability to generalize across structurally distinct tasks and sustain reasoning over extended sequences.
>
> ## Compatibility with Existing LLMs
> Beyond being a novel benchmark, AnyMDP bridges a critical gap in assessing experience-driven, holistic reasoning capabilities of pre-trained LLMs. Existing benchmarks predominantly measure declarative memory and one-shot circuits, while AnyMDP emphasizes procedural memory and experience-driven learning.
>
> ## Low-cost Access to Oracle Policy
> This property significantly enhances the efficiency and scalability of ICRL training, making AnyMDP a scalable benchmark.
>
> ## Empirical analysis across state-of-the-art architectures
> We also augment the empirical analysis with the training curves and ICL comparisons across state-of-the-art architectures, including Mamba2, RWKV-7, Gate Delta Net, and other competitive models. This will demonstrate AnyMDP’s utility as a rigorous benchmark for advancing long-context modeling capabilities.
>
> - Table 1. Validation Loss of Different Models on AnyMDP Datasets
> | CL 	  | 1  	   | 100	    | 1000    | 10000   | 30000 |
> |-------------|------------|------------|------------|------------|------------|
> |  Mamba2   | 1.417 ± 0.035 | 1.272 ± 0.051 | 0.822 ± 0.064 | 0.436 ± 0.053 | 0.294 ± 0.042 |
> |  GSA   | 1.406 ± 0.034 | 1.235 ± 0.053 | 0.765 ± 0.066 | 0.342 ± 0.050 | 0.235 ± 0.042 |
> |  Gate DeltaNet   | 1.356 ± 0.035 | 1.130 ± 0.051 | 0.599 ± 0.055 | 0.267 ± 0.049 | 0.173 ± 0.042 |
> |  RWKV-7   | 1.371 ± 0.034 | 1.151 ± 0.052 | 0.627 ± 0.055 | 0.269 ± 0.042 | 0.175 ± 0.029 |
>
>
> Overall, in our research on ICL, the benefits brought by the discrete MDPs far outweigh the drawbacks of its limited problem space. We have also started to explore ICL in the continuous space.
>
> For future work in continuous settings, there are two directions. First, we will focus on specific continuous problems and prioritize in-domain adaptation. For instance, in frame prediction tasks, generalization manifests as adaptation to environments with distinct features and spatial structures. Similarly, in robot whole-body locomotion control, it manifests in adapting to different robot configurations (size, shape, weight). Second, we will continue exploring how to generate tasks with high diversity and low structural bias in continuous spaces. Undoubtedly, the first direction holds more practical value, while the second poses greater theoretical challenges.
>
> # Explanation of AnyMDP
> ## How to generate high-quality MDPs
> (To answer the reviewer's Weakness 1 and Question 1.)
>
> The sampling of transition matrix and reward matrix in AnyMDP is not uniformly random, to avoid generating trivial MDP problems (e.g., MDPs with optimal episode length=1). Uniform randomization would result in generating a majority of trivial Markov Decision Processes (MDPs), such as those with optimal episode lengths of 1 or MDPs where most states are irrelevant to the optimal solution. The policy only needs to remember which short path has the best reward and will not learn the trade-off of risk assessment. Eventually, the environment will be more inclined towards the bandit problem. In most complex RL problems, delayed reward is quite common. Usually, a large positive reward is only obtained when the goal is achieved. Sometimes, there may even be a stage of deduction before getting the positive reward.
>
> Ergodicity is ensured by Theorem 1 in the appendix, which guarantees that a uniformly random policy induces a strictly positive probability of reaching every state from the initial state. At the same time, AnyMDP does not guarantee that the optimal policy is strictly ergodic—a trait absent in most existing benchmarks. However, the banded transition kernel and reward structure ensure that the optimal policy requires transitions across a substantial number of states.
>
> As we have presented above, AnyMDP can serve as a Long-Term Dependency Benchmark. For this reason, we must limit the step length—meaning that to reach high-value targets, the path cannot be short. As the number of states increases, the path to achieving a high score grows correspondingly longer. Thus, "Under a uniform random policy, the probability of reaching high-valued states remains greater than 0 but decreases at least exponentially with respect to $n_s$."
>
> ## Behavior policies
> (To answer the reviewer's Weakness 3.)
>
> On page 20, we present Table 5 about "Correspondence of prompt IDs and the policies it represents". We thank the reviewer for pointing out the lack of clarity in this part, and will subsequently include them in the paper.
> - For IDs 0 to 3, the policies are value iteration policies with different gamma factors, corresponding to short-term to long-term oracles ($\gamma$=0.5, 0.93, 0.994).
> - The model-based reinforcement learner comprises two distinct components: an environment model and a reward model, both conditioned on the current interaction history. A Bellman update is applied at every step to compute the mean value function, which is subsequently augmented with Optimistic Thompson Sampling to balance exploration and exploitation.
> -  Tabular Q-learner with UCB is a model-free method that maintains a Q-table of state-action values. These values are iteratively refined via value iteration and adjusted by an upper-confidence-bound bonus derived from rewards and successor-state information observed during interaction.
>
> Regarding the exact impact of the diversity of behavior policies in training data, DPT has demonstrated its benefit on model performance. Indeed, compared with DPT, our primary contribution lies in proposing the Step-wise Supervision method, which enhances the strength of supervisory signals and improves training efficiency. As for policy tags, they serve as an additional benefit—analogous to high-quality prompts in LLMs—by informing the model of the actual level/type of the current action, thereby facilitating strategy distillation during training.
>
> # Gymnasium environments and reward shaping
> (To answer the reviewer's Question 2.)
>
> We select the Gym envs based on two points: 1. The env should have a delayed reward to make it more difficult. 2. The observation and action space can match our model. We will also cite the Deep Sea environment, and compare the difference with the Gymnasium.
>
> (To answer the reviewer's Question 3.)
>
> Reward shaping is a common practice in reinforcement learning, and it is applied consistently across all baselines as well as our model—we consider this approach fair. The quality of such reward shaping is analogous to that of prompts in large models: it helps render the problem more intuitive. Notably, without reward shaping, both the context size required for our model’s online learning and the number of training iterations needed for baselines would increase.

---

> > ### Comment · Reviewer_WkeX · 2025-08-07
> >
> > Dear authors, thank you for the additional explanations, in particular on how you define high-quality MDPs. Regarding the reward shaping, some more extensive discussion regarding its impact would be appreciated in the paper.

---

> > > ### Author Response · Authors · 2025-08-08
> > >
> > > We greatly appreciate the reviewers for their constructive comments and perceptive suggestions. We will further explain the meaning of the reward shaping design in Figure 12 on page 23:
> > > - For Frozen Lake, the original reward schedule specifies that reaching the goal gives +1, while reaching a hole or frozen area gives 0. We consider that falling into a hole, as a failure scenario, should be penalized, so we adjusted the reward for falling into a hole to -1. This is equivalent to giving the model a prompt: reach the goal under the safest conditions. Additionally, in the non-slippery setting, Frozen Lake becomes a navigation problem where the optimal strategy is to reach the goal as quickly as possible without falling into holes; thus, we added a step-wise cost, meaning that reaching a frozen area yields a penalty of -0.05. At this point, the prompt becomes: reach the goal as quickly as possible.
> > > - For Cliff Walking, the original reward is such that each time step yields -1, unless the player steps onto the cliff, in which case it yields -100. Since the rewards in our training data range from -1 to 1, a reward of -100 is excessively large and thus requires normalization—this is also common in RL algorithms. For example, in Stable Baselines3, the VecNormalize tool can be used for such numerical processing.
> > > - For Pendulum, $r = -\left (\theta^2 + 0.1 \cdot \theta_{\text {dt}}^2 + 0.001 \cdot \text {torque}^2 \right)$. First, we performed value normalization, and then we clipped the excessively low rewards. The purpose of the Pendulum environment is to upright the pole, which requires first swinging left and right at the bottom to accelerate and gain kinetic energy—this means it must first go through a period where the reward gradually decreases. To encourage the algorithm to explore, we set a lower bound for the reward, and this is the reason for the clipping operation.
> > > - For Switch, the original reward scheme is as follows: each agent receives a reward of +5 for reaching its home cell. The episode terminates when both agents have reached their home cells or upon hitting a maximum of 100 steps in the environment. In addition to value normalization, we added rewards to each agent: +0.08 for approaching the target, -0.12 for moving away from the target, and -0.04 for remaining stationary. We then combined the rewards of all agents and provided them to the agents. The purpose of this design is to clarify that there is a reward for approaching the target while introducing a short-term reward bias—if both agents act greedily, neither will reach the goal. Surprisingly, we found that one of the agents was able to use the most greedy approach to reach the goal in the minimum number of steps, while the other agent learned to give way first, adopting a strategy of sacrificing itself in the short term for the greater good.

---

### Official Review · Reviewer_nuE2 · 2025-07-02

**Clarity:** 1
**Significance:** 3
**Originality:** 3
**Rating:** 3
**Confidence:** 3

**Summary:**

The paper studies the in-context reinforcement learning (ICRL) problem. It proposes AnyMDP, a class of procedurally generated tabular MDPs. Additionally, it proposed to scale meta-training via step-wise supervision and prior knowledge in the form of action markers corresponding to the behavior policy used. Through this, the paper empirically demonstrates that when the number of AnyMDP tasks used during meta-training is large, the model can generalize to unseen tasks.

**Questions:**

1. Section 4.2 demonstrates the effectiveness of incorporating prior information (action markers). Do the AD and DPT baselines have access to this information too? How realistic is the access to this prior information? Also, do the authors have any intuition as to why it helps significantly?
2. The unseen tasks in Section 4.3 are basically from the AnyMDP class, and hence, in-distribution. Isn't generalization in this setting sort-of guaranteed?

**Ethical Concerns:**

["NO or VERY MINOR ethics concerns only"]

**Final Justification:**

I would like to maintain my score as my concerns remain unaddressed.

**Limitations:**

Yes.

**Paper Formatting Concerns:**

None.

**Quality:**

2

**Strengths And Weaknesses:**

Disclaimer: I'm not deeply familiar with ICRL literature, so please take my comments with that in mind.

**Strengths**
- The paper studies ICRL, which is a problem of current interest to the community.
- Some empirical results are quite strong (e.g., models trained on AnyMDP tasks generalizing to (discretized) Pendulum from Gymanisum).

**Weaknesses**
- My major concern with this paper is the writing. As someone who is not particularly familiar with the ICRL literature, the paper felt a bit hard to follow at times. For instance, in-context learning is not formally defined anywhere, unusual notations: \tau for MDPs, {Causal}_{\theta} defined nowhere, hard to understand what Fig. 1 is trying to convey, etc.
- Garnet MDPs is a widely used class of procedurally generated MDPs in the RL community [1, 2, 3]. The paper neither mentions nor compares AnyMDPs with Garnet MDPs.
- While the paper claims many results on unseen tasks (e.g., Section 4.3), all of the test tasks considered in those setups are in-distribution, which is not particularly interesting. The results showcasing generalization on Gymnasium tasks (out of distribution) are interesting.
- (Minor) At times, the paper would reference a figure in the main results but the figure would be in the appendices (e.g., Fig. 15, line 6) . This is not a good writing habit.

Overall, the paper studies an interesting problem and provides some good results. However, the writing requires a lot more polish (along with proper comparison with prior work on procedurally generating MDPs) to be apt for publication at NeurIPS.

[1] Bhatnagar et al. Natural Actor–Critic Algorithms, Automatica (Journal of IFAC) 2009.

[2] Castro et al. MICo: Improved representations via sampling-based state similarity for Markov decision processes, NeurIPS 2021.

[3] Castro et al. A Kernel Perspective on Behavioural Metrics for Markov Decision Processes, TMLR 2023.

---

> ### Author Rebuttal · Authors · 2025-07-29
>
> First and foremost, we sincerely thank the reviewer for identifying the typos and providing valuable suggestions on the writing. We will revise accordingly to further improve the paper’s readability, especially on the explanation of ICL and Casual Blocks. In the following, we will elaborate in detail on the weaknesses and questions raised by the reviewer.
>
> # Garnet MDP and other related works
> We appreciate the reviewer’s reference to Garnet MDP, an efficient MDP sampling methodology. Key parallels and distinctions include:
> - Similarities:
>      - Both prioritize MDP structural integrity
>      - Both focus on stopping "fast mixing" MDPs. (In our case, theorem 1 also guarantees this point)
> - Differences:
>     - Garnet MDP controls the transitions under an optimal policy, whereas AnyMDP controls the transitions under the uniformly random policy. In our case, we guarantees that a uniformly random policy induces a strictly positive probability of reaching every state from the initial state (ergodic). However, AnyMDP does not guarantee that the optimal policy is strictly ergodic---a better property aligns with most existing benchmarks. Nonetheless, the banded transition kernel and reward structure ensure that the optimal policy requires transitions across at least a substantial number of states (nearly ergodic).
>     - Garnet MDP is tailored for infinite-horizon MDPs, while AnyMDP supports both infinite- and finite-horizon settings (aligning with common RL benchmarks).
>
> The paper, Natural actor–critic algorithms, presents four new natural actor–critic reinforcement learning algorithms, provides their convergence proofs, and improves upon prior work by being the first to offer such proofs and propose fully incremental algorithms for these methods. MICo, the learnt representations in MDP problems, demonstrated a gratifying performance improvement for DQN and SAC deep reinforcement learning agents. A Kernel Perspective on Behavioural Metrics defines a new metric equivalent to the reduced MICo distance, establishes new theoretical results, and demonstrates strong empirical effectiveness in deep reinforcement learning.
>
> While our work also focuses on MDP problems, the emphasis differs. The AnyMDP we propose aims to generate tasks with both diversity and quality. Using datasets generated by AnyMDP, we train a generalizable OmniRL model to investigate the in-context learning (ICL) mechanism of large models.
>
> We appreciate the reviewer's suggestion. We will include a proper comparison with prior work on procedurally generating MDPs in the related works section and cite these papers.
>
> # AD, AD𝜖, DPT, and our AnyMDP
> ## Explanation of Figure 1
> Figure 1 illustrates the differences between AD, DPT, and our AnyMDP. Blue arrows (Behavior) depict the trajectory fed to the model; yellow arrows (Reference) provide the corresponding supervisory signal. (Left) For AD, the reference action is identical to the behavioral action. (Middle) For a sequence of behaviors, DPT provides a reference action only at the end of the sequence. (Right) For our method, we provide a reference action at each step of the behavioral sequence, which in turn enhances the strength of the supervisory signal and improves training efficiency. The key contribution of Stepwise Supervision is that it tolerates step-level inconsistencies between behavior and reference, aiming at solving the issue of distribution shift in imitation learning.
> ## Explanation of prior information
> The reviewers raised questions regarding the inclusion of prior information in AD, AD𝜖, and DPT. To clarify, prior information introduces additional details about different behavioral policies—specifically, policy tags. Since the AD policy remains unchanged during generation, adding prior information has no theoretical effect. For AD𝜖, policy tags can be incorporated based on the noise level. Additionally, as DPT involves multiple policies, the inclusion of policy tags is feasible. We believe that adding policy tags is also highly likely to enhance performance for AD𝜖 and DPT.
>
> Indeed, compared with DPT, our primary contribution lies in proposing the Stepwise Supervision method. As for policy tags, they serve as an additional benefit—analogous to high-quality prompts in LLMs—by informing the model of the actual level/type of the current action, thereby facilitating strategy distillation during training.
>
> The prior information is accessible. During the training phase, the classification of prior information is manually designed, and we randomly mask some tags as "Unknown". In online testing, we allow the model to assess current performance with all tags set to "Unknown".
>
> # Experiment settings in Section 4.3
> Section 4.3 aims to examine how the number of tasks influences generalization. While all tasks in our experiments are drawn from AnyMDP, we ensured sufficient diversity among them—and our results support this observation. Specifically, models trained on fewer tasks showed limited performance on other AnyMDP tasks, whereas satisfactory generalization to additional AnyMDP tasks was only achieved when the number of training tasks was sufficiently large. This aligns with the key point we intended to convey: a larger number of tasks can foster generalization.
>
> Furthermore, our experimental observations suggest that if a model fails to generalize within AnyMDP, it is unlikely to generalize to Gym tasks either. We believe that out-of-distribution generalization is inherently more challenging than in-distribution generalization.

---

> > ### Comment · Reviewer_nuE2 · 2025-08-07
> >
> > I thank the authors for the rebuttal.
> >
> > “Garnet MDP controls the transitions under an optimal policy” — could the authors explain this claim?
> >
> > “We believe that adding policy tags is also highly likely to enhance performance for AD𝜖 and DPT.” — then a fair comparison is possible only if these tags are incorporated. What if the gain is due to the policy tags alone?

---

> ### Author Response · Authors · 2025-08-08
> **About the Follow-Up Questions**
>
> We sincerely appreciate the reviewer’s insightful feedback, and we carefully address the reviewer's concern as below. We will also include the necessary discussion and revision in the paper to answer those questions.
>
> ## Q1: Compare AnyMDP with Existing Benchmarks
> - (Archibald et al., 1995) outlines a three-step procedure for constructing MDPs: (1). Transition structure generation for the optimal policy,(2). Finalization of reward and transition data for the optimal policy (ensuring ergodicity), and (3). Incorporation of non-optimal policy connections. It prevents "fast mixing" of states and also ergodicity of state-transition under optimal policy. Theorem 1 in our paper establishes two key properties under a uniform random policy: (1). It precludes fast mixing (i.e., the stationary distribution decay at least exponentially), ensuring convergence is not overly rapid, which is critical for stable learning dynamics. (2). It guarantees ergodicity (the stationary distribution is strictly positive for all states), ensuring all states are reachable under this policy. An analysis of existing benchmarks reveals a critical trade-off: while transitions under a uniformly random policy are designed to be ergodic (as required for exploratory robustness), transitions under an optimal policy are not inherently ergodic.
>
> - The Garnet MDP (Bhatnagar et al., 2007) represents another class of MDP samplers, parameterized as Garnet(n, m, b, σ, τ). Notably, Garnet supports non-stationary MDPs when τ != 0, with parameter b constraining state-action pair transitions. Upon closer examination—and excluding cases with absorbing states—**AnyMDP can be interpreted as a subset of Garnet(n_s, n_a, b = b₋ + b₊, σ)** （ b₋ , b₊ are the parameters in Algorithm1 in appendix）. However, Garnet does not inherently guarantee ergodicity (e.g., unreachable states may exist) or fast mixing, and more closely aligns with a variant of AnyMDP that lacks composite rewards and banded transition. In contrast, AnyMDP emphasizes irreducibility, longer-delayed rewards, and more challenging tasks, distinguishing it from standard Garnet formulations.
>
> Importantly, neither the Archibald et al. (1995) framework nor Garnet MDPs explicitly **account for absorbing states (e.g., terminal states requiring environment resets, such as pitfalls or goal states)**. This could also be a crucial limitation to generalize to existing benchmarks.
>
>
> ## Q2: The contribution of prior (tag) and stepwise supervision
> The core distinction between DPT and Stepwise Supervision (SS) lies in their **data structuring and training efficiency**
>
> - DPT structures data as **<Trajectory, Independently Sampled Single Query State, Reference Action>** tuples
> - Stepwise Supervision is theoretically identical to **<Trajectory, Sequence of All Trajectory States as Query States, Sequence of Reference Actions>** tuples.
>
> For each trajectory, DPT trains only one state-action pair, whereas Stepwise Supervision leverages all state-reference action pairs across the trajectory. This makes Stepwise Supervision theoretically **L-fold more training-efficient** (where L = trajectory length). In our context (context length over 8K to 512K, DPT becomes extremely in efficient.
>
> Our revised manuscript will emphasize that Stepwise Supervision offers significantly greater scalability and training efficiency compared to DPT. We will introduce the DPT+ prior in the final version to address these limitations. Figure 6 directly contrasts DPT with OmniRL (without prior knowledge), effectively isolating and demonstrating SS's advantages over DPT.

---

### Official Review · Reviewer_1krJ · 2025-07-07

**Clarity:** 4
**Significance:** 3
**Originality:** 4
**Rating:** 5
**Confidence:** 4

**Summary:**

* Proposes to procedurally generate tabular MDPs in order to study various properties of ICRL
* Results demonstrate that with a sufficiently large scale of tasks, the proposed model can generalize to tasks unseen in training set

**Questions:**

* Which findings do the authors believe would(n't) scale to more complex domains?
* How might future work approach the challenge of broadening the scope of the efforts in this work to continuous settings, and ones where procedural generation may not be possible?

**Ethical Concerns:**

["NO or VERY MINOR ethics concerns only"]

**Final Justification:**

I stand by my initial review and therefore maintain my current score.

**Limitations:**

Yes

**Quality:**

4

**Strengths And Weaknesses:**

Strengths
* The paper is well-written and clearly presented; in fact, exceptionally so
* The details and analysis in the main body are thorough but not overwhelming
* OmniRL method produces very strong performance against baselines
* Useful insights---especially that "generalization of ICRL can be at odds with its zero-shot or few-shot performance"

Weaknesses
* Only applicable to discrete state and action spaces (acknowledged as limitation)
* Procedural generation is difficult to scale to more complex domains; but I consider the toyishness of this work a strength, as it enabled in-depth analysis (hard to do in non-tabular settings, where PCG is not possible); and I personally believe it is likely the findings would be similar for more complex domains (if such thorough experiments could be run); but see related questions below

---

> ### Author Rebuttal · Authors · 2025-07-29
>
> # On the Value of Discrete Space MDPs
> We appreciate the reviewer's positive feedback on the value of the seemingly toyishness of the discrete MDPs.  At this stage, partially pursuing the task complexity risks undermining the core advantages of procedurally generated discrete MDPs. Below, we explain the challenges of preserving key properties in continuous domains and contrast our approach with contemporary benchmarks.
>
> ## Challenges of Continuous MDPs in Preserving Structural Integrity
> Extending procedurally generated MDPs to continuous spaces while maintaining low structural bias and sufficient diversity poses significant technical hurdles. At the current stage, efforts moving to continuous MDPs risk typically **prioritizing "absolute task complexity" over qualitative diversity**, introducing biases or homogenizing task structures—contrary to our framework's goals.
>
>
> ## Inherent Long-Term Dependency and Qualitative Diversity
> Existing continuous MDP benchmarks rely on continuously randomized hyperparameters to simulate "infinite tasks," prioritizing few-shot learning over long-horizon in-context learning (ICL). Our analysis reveals two insights:
> - Task Diversity ≠ Numerical Scale: True diversity depends on qualitative variation in task structures, not just quantity.
> - ICL efficacy depends on disparity between tasks in the training set, not individual task complexity. Existing continuous benchmarks tend to produce homogeneous task structures, limiting models to superficial pattern recognition via in-weight learning (IWL) rather than sustained reasoning. In contrast, discrete MDPs in AnyMDP are designed with procedural complexity and temporal coherence, enabling rigorous evaluation of a model’s ability to generalize across structurally distinct tasks and sustain reasoning over extended sequences.
>
> ## Compatibility with Existing LLMs
> Beyond being a novel benchmark, AnyMDP bridges a critical gap in assessing experience-driven, holistic reasoning capabilities of pre-trained LLMs. Existing benchmarks predominantly measure declarative memory and one-shot circuits, while AnyMDP emphasizes procedural memory and experience-driven learning.
>
> ## Low-cost Access to Oracle Policy
> This property significantly enhances the efficiency and scalability of ICRL training, making AnyMDP a scalable benchmark.
>
> ## Empirical analysis across state-of-the-art architectures
> We also augment the empirical analysis with the training curves and ICL comparisons across state-of-the-art architectures, including Mamba2, RWKV-7, Gate Delta Net, and other competitive models. This will demonstrate AnyMDP’s utility as a rigorous benchmark for advancing long-context modeling capabilities.
>
> - Table 1. Validation Loss of Different Models on AnyMDP Datasets
> | CL 	  | 1  	   | 100	    | 1000    | 10000   | 30000 |
> |-------------|------------|------------|------------|------------|------------|
> |  Mamba2   | 1.417 ± 0.035 | 1.272 ± 0.051 | 0.822 ± 0.064 | 0.436 ± 0.053 | 0.294 ± 0.042 |
> |  GSA   | 1.406 ± 0.034 | 1.235 ± 0.053 | 0.765 ± 0.066 | 0.342 ± 0.050 | 0.235 ± 0.042 |
> |  Gate DeltaNet   | 1.356 ± 0.035 | 1.130 ± 0.051 | 0.599 ± 0.055 | 0.267 ± 0.049 | 0.173 ± 0.042 |
> |  RWKV-7   | 1.371 ± 0.034 | 1.151 ± 0.052 | 0.627 ± 0.055 | 0.269 ± 0.042 | 0.175 ± 0.029 |
>
>
> Overall, in our research on ICL, the benefits brought by the discrete MDPs far outweigh the drawbacks of its limited problem space. We have also started to explore ICL in the continuous space.
>
> # Question about more complex domains and continuous settings
> - For future work in continuous settings, there are two directions. First, we will focus on specific continuous problems and prioritize in-domain adaptation. Second, we will continue exploring how to generate tasks with high diversity and low structural bias in continuous spaces. Undoubtedly, the first direction holds more practical value, while the second poses greater theoretical challenges.
> - If the second direction remains unaddressed, cross-domain generalization in continuous spaces will be difficult to achieve, but in-domain adaptation is highly feasible. For instance, in frame prediction tasks, generalization manifests as adaptation to environments with distinct features and spatial structures. Similarly, in robot whole-body locomotion control, it manifests in adapting to different robot configurations (size, shape, weight).
> - Another more complex domain is **Multi-agent** problem. As agents need to collaborate and their performance cannot be guaranteed to be consistent, a new challenge arises: in-context adaptation across other agents' actions. If a single agent can observe and learn other agents' actions online, while adjusting its own real-time policy, it could potentially alleviate the fragility of existing multi-agent systems. We believe this scenario will uncover more unknown findings.

---

### Note · Authors · 2025-08-12

Regarding the validation of scalability claims for larger problems, we would like to highlight key evidence demonstrating AnyMDP’s inherent scalability: 1.Scalable Context Length: OmniRL is designed to handle large contexts (over 10K tokens), with practical scalability demonstrated by expanding state space representation to increase context length (up to 40K tokens) and state dimensionality (up to 128 states). This exceeds most LLM benchmarks, and the main limitation is the model's memory retention, not the benchmark design. 2.Scalable Task Complexity: Empirical evidence in Figure 6 shows that increasing tasks (from 10K to 128K) improves ICRL training performance, even with a modest state space size (16). This task scale is among the largest reported in ICRL literature, and further scaling is expected with larger state spaces.

We performed a quantified comparison of AnyMDP and Garnet, analyzing their stationary distributions under random and oracle policies. As established in Theorem 1, AnyMDP demonstrates exponential decay in transition probabilities under random policies.
| Transitions|  State 1 | State 8 | State 32 |
|-------|-------|-------|---------|
| AnyMDP(64,5) - Random Policy | 0.127  | 0.043 | 0.004 |
| Garnet(64,5,2) - Random Policy | 0.032  | 0.023 | 0.015 |
| Garnet(64,5,4) - Random Policy | 0.027  | 0.021 | 0.016 |
| AnyMDP(64,5) - Oracle Policy | 0.204  | 0.038 | 0.001 |
| Garnet(64,5,2) - Oracle Policy | 0.129  | 0.038 | 0.003 |
| Garnet(64,5,4) - Oracle Policy | 0.093  | 0.033 | 0.011 |

We also evaluated TQL-UCB, PPO, and OmniRL on Garnet MDPs. AnyMDP proved more challenging than Garnet for PPO and TQL, reflecting its greater complexity. Despite notable differences between the task distributions, OmniRL trained on AnyMDP showed limited but meaningful generalization.

| Tasks | TQL-UCB | PPO | OmniRL (Trained with AnyMDP Only)  |
|-------|-------|-------|---------|
| AnyMDP(16,4) | 92.0\%/297K/4.7K | 90.6\%/476K/9.7K | 95.3\%/2.0K/29$ |
| AnyMDP(64,4) |83.7\%/1.1M/5.1K | 58.3\%/1.1M/9.4K | 91.3\%/7.7K/25 |
| Garnet(16,5,2,0.5) | 98.8%/241K/2.1K  | 97.1%/57K/0.5K | 86.3%/8.2K/71  |
| Garnet(64,5,2,0.5)|  98.7%/614K/1.7K |  98.1%/96K/0.26K |80.4%/8.0K/19 |

Finally, we sincerely appreciate the insightful feedback and constructive suggestions from all the reviewers.

---

### Decision · Program_Chairs · 2025-09-17

**Decision:**

Accept (poster)

**Comment:**

This paper introduces the AnyMDP framework, which leverages randomized MDPs to generate diverse tasks for large-scale meta-training in In-Context Reinforcement Learning (ICRL). By incorporating step-wise supervision and prior knowledge induction, the framework improves both efficiency and generalization. Experiments show that OmniRL, trained on these randomized tasks, achieves strong performance on unseen environments. The work highlights the importance of task diversity and world randomization in scaling ICRL and improving robustness, with results demonstrating that larger task sets enhance generalization, albeit with increased adaptation time.

All reviewers agree that the paper makes novel and valuable contributions, particularly in enabling scalable task generation and reducing structural bias through world randomization. Concerns remain regarding task complexity and presentation. While the writing is generally clear and well organized, the explanation of key results—especially in figure captions—could be improved. Moreover, the proposed task generation does not yet scale to complex or continuous domains, and these limitations should be explicitly acknowledged in the camera-ready.